# Epidermal threads reveal the origin of hagfish slime

Yu Zeng[1]*, David C Plachetzki[2], Kristen Nieders[1], Hannah Campbell[1], Marissa Cartee[2,3], M Sabrina Pankey[2], Kennedy Guillen[1], Douglas Fudge[1]

[1]Schmid College of Science and Technology, Chapman University, Orange, United States; [2]Department of Molecular, Cellular and Biomedical Sciences, University of New Hampshire, Durham, United States; [3]Department of Evolution, Ecology and Organismal Biology, University of California at Riverside, Riverside, United States

**Abstract** When attacked, hagfishes produce a soft, fibrous defensive slime within a fraction of a second by ejecting mucus and threads into seawater. The rapid setup and remarkable expansion of the slime make it a highly effective and unique form of defense. How this biomaterial evolved is unknown, although circumstantial evidence points to the epidermis as the origin of the thread- and mucus-producing cells in the slime glands. Here, we describe large intracellular threads within a putatively homologous cell type from hagfish epidermis. These epidermal threads averaged ~2 mm in length and ~0.5 μm in diameter. The entire hagfish body is covered by a dense layer of epidermal thread cells, with each square millimeter of skin storing a total of ~96 cm threads. Experimentally induced damage to a hagfish's skin caused the release of threads, which together with mucus, formed an adhesive epidermal slime that is more fibrous and less dilute than the defensive slime. Transcriptome analysis further suggests that epidermal threads are ancestral to the slime threads, with duplication and diversification of thread genes occurring in parallel with the evolution of slime glands. Our results support an epidermal origin of hagfish slime, which may have been driven by selection for stronger and more voluminous slime.

*For correspondence: dreavoniz@berkeley.edu

Competing interest: The authors declare that no competing interests exist.

## Editor's evaluation

The study is a careful investigation of the physical properties of hagfish slime and the underlying cellular framework that enables this extraordinary evolutionary innovation. It is a careful and detailed measurement with clear images. The revised manuscript provides a better contextualizing of the findings as a broader biological question, including the evolution of functional novelty, the adaptive processes, and the links between genetic and phenotypic evolution. The transcriptome analysis of several species further supports the evolutionary model. Therefore, this paper provides solid evidence for a unique and important view of the slime and should be of interest to those working on hagfish and on these secretions.

## Introduction

Among the various defensive structures used by animals, hagfish slime is a fibrous hydrogel that is recognized for its exceptional material properties and unique deployment mechanisms (*Ewoldt et al., 2011*; *Chaudhary et al., 2018*; *Lim et al., 2006*; *Winegard and Fudge, 2010*; *Bernards et al., 2018*). The hagfishes (class Myxini) are a group of epibenthic jawless vertebrates possessing a row of slime glands along each side of their body, with each gland producing and storing gland thread cells (GTCs) and gland mucous cells. When a hagfish is attacked, it produces defensive slime by rapidly ejecting ruptured gland mucous cells and GTCs into seawater (*Figure 1A*). Within 400 ms after ejection, coiled

**eLife digest** Hagfishes are deep-sea animals, and they represent one of the oldest living relatives of animals with backbones. To defend themselves against predators, they produce a remarkable slime that is reinforced with fibers and can clog a predator's gills, thwarting the attack. The slime deploys in less than half a second, exuding from specialized glands on the hagfish's body and expanding up to 10,000 times its ejected volume. The defensive slime is highly dilute, consisting mostly of sea water, with low concentrations of mucus and strong, silk-like threads that are approximately 20 centimeters long. Where and how hagfish slime evolved remains a mystery.

Zeng et al. set out to answer where on the hagfish's body the slime glands originated, and how they may have evolved. First, Zeng et al. examined hagfishes and found that cells in the surface layer of their skin (the epidermis) produce threads roughly two millimeters in length that are released when the hagfish's skin is damaged. These threads mix with the mucus that is produced by ruptured skin cells to form a slime that likely adheres to predators' mouths. This slime could be a precursor of the slime produced by the specialized glands. To test this hypothesis, Zeng et al. analyzed which genes are turned on and off both in the hagfishes' skin and in their slime glands. The patterns they found are consistent with the slime glands originating from the epidermis.

Based on these results, Zeng et al. propose that ancient hagfishes first evolved the ability to produce slime with anti-predator effects when their skin was damaged in attacks. Over time, hagfishes that could produce and store more slime and eject it actively into a predator's mouth likely had a better chance of surviving. This advantage may have led to the appearance of increasingly specialized glands that could carry out these functions.

The findings of Zeng et al. will be of interest to evolutionary biologists, marine biologists, and those studying the ecology of predator-prey interactions. Because of its unique material properties, hagfish slime is also of interest to biophysicists, bioengineers and those engaged in biomimetic research. The origin of hagfish slime glands is an interesting example of how a new trait evolved, and may provide insight into the evolution of other adaptive traits.

threads from GTCs unravel and mucous vesicles from gland mucous cells swell and deform, resulting in a network of mucus and threads that entraps large volumes of water and effectively clogs the mouth and gills of fish predators (*Lim et al., 2006*; *Zintzen et al., 2011*). A single pinch on the tail of an adult Pacific hagfish (*Eptatretus stoutii*; ~45 cm body length) can cause the production of 0.9 l of slime, which is ~7 times the volume of the animal (~0.14 l) (*Fudge et al., 2005*). There is no other biological or synthetic material that can expand so much in so little time.

Despite being composed mostly of water (i.e., >99.99% seawater), hagfish slime is strong and viscoelastic (*Fudge et al., 2003*; *Fudge et al., 2005*; *Ewoldt et al., 2011*; *Fudge et al., 2015*; *Böni et al., 2016*). The impressive strength of the slime is imparted by a network of slime threads, whose diameter and length vary from 0.7 to 4 µm and 5–22 cm, respectively (*Zeng et al., 2021*). These threads consist mainly of fibrous α and γ proteins from the intermediate filament family, and they are individually produced and stored within GTCs as a densely packed skein (*Downing et al., 1981*; *Downing et al., 1984*; *Spitzer et al., 1988*; *Koch et al., 1995*; *Winegard et al., 2014*). As a strong biopolymer, they rival spider silk in their strength and toughness and are the largest cytoplasmic polymers known in biology (*Fudge et al., 2003*; *Fudge and Gosline, 2004*; *Fudge et al., 2010*; *Zeng et al., 2021*).

Defensive sliming in hagfishes can be considered a key innovation for this group, but its evolutionary origin remains unclear. The slime renders hagfishes essentially free from predation by gill-breathing predators, which may explain how this group has persisted while most other jawless fishes have gone extinct since the rise of jawed vertebrates (*Randle and Sansom, 2019*). The fossil record for hagfishes is sparse, which makes tracing the evolutionary origins of the slime glands difficult. The youngest hagfish fossil without slime glands dates to the Pennsylvanian period (~310 million years ago; *Miyashita, 2020*), while the oldest hagfish fossil with slime glands dates to the Cretaceous (~138 million years ago; *Miyashita et al., 2019*), suggesting a wide window of time when these structures could have evolved (*Figure 1B*). Other evidence suggests an even earlier origin of the threads. Lampreys, which are the closest living relatives of the hagfishes, also possess a kind of thread

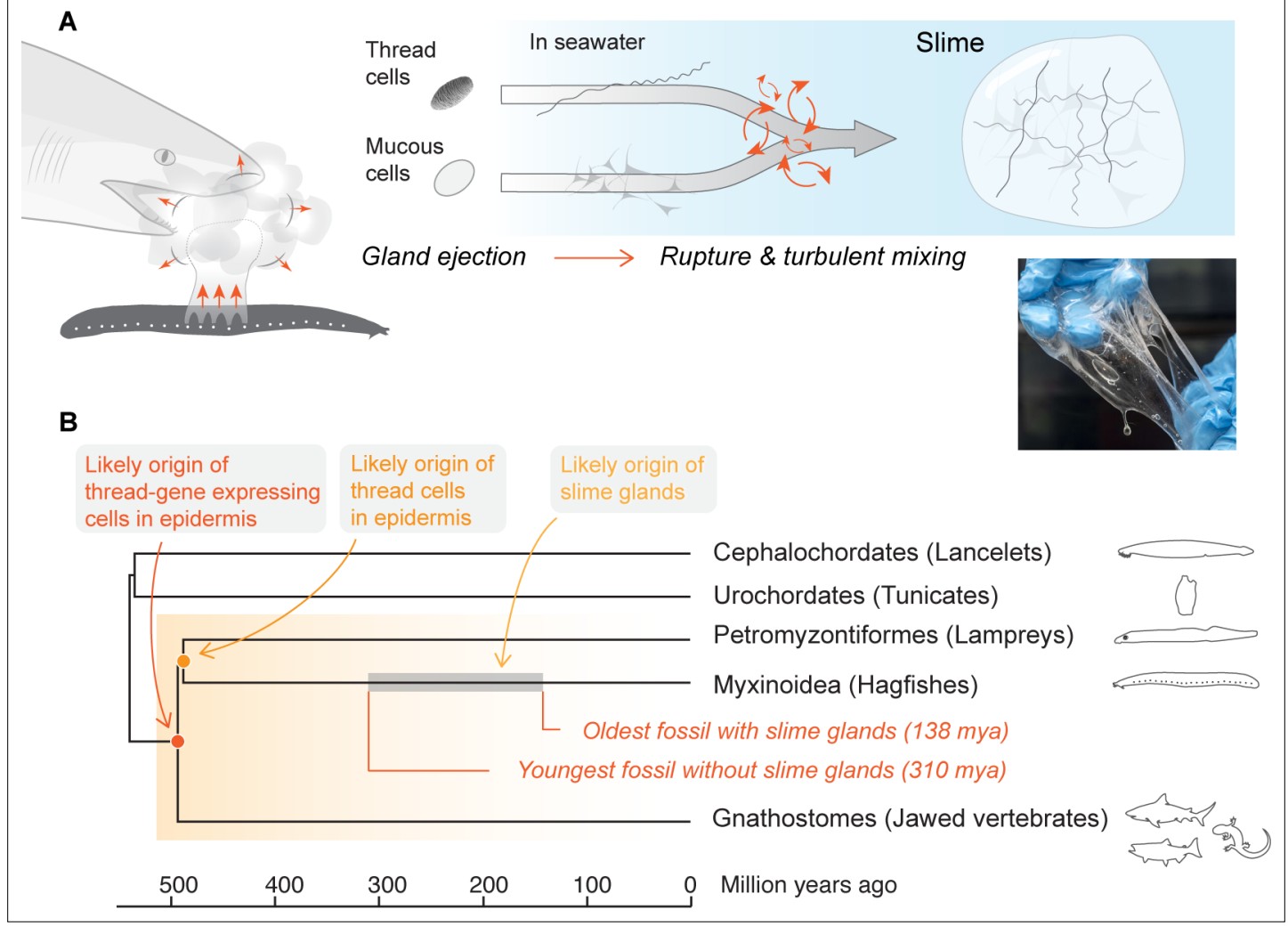

**Figure 1.** Mechanism and evolutionary history of hagfish slime. (**A**) Hagfish defensive slime is produced by rapid ejection and rupture of mucous cells and thread cells into seawater by slime glands. Top shows a schematic sequence of slime formation. Threads and mucus are released from ruptured cells and mix with seawater to form large volumes of dilute, soft, viscoelastic slime (lower right). (**B**) A consensus tree of chordates highlighting the origins of epidermal thread cells and hagfish slime glands. Orthologs of intermediate filament thread genes are expressed in the skin of lampreys, hagfishes, teleost fishes, and amphibians (orange shade), and thus likely have an origin that dates back to the common ancestor of vertebrates. Epidermal cells producing large threads are only known in hagfishes and lampreys (see also *Figure 1—figure supplement 1*), and thus epidermal thread cells likely originated in their common ancestor. The gray segment highlights a wide window of time between 138 and 310 million years ago when the hagfish slime glands evolved (*Miyashita, 2020*).

The online version of this article includes the following figure supplement(s) for figure 1:

**Figure supplement 1.** Cross-section of lamprey (*Petromyzon marinus*) epidermis based on hematoxylin-eosin (H&E)-stained cross-sectional slides.

cell in their epidermis called a 'skein cell' (*Lane and Whitear, 1980*; also see below). Also, the genes closely related to the α and γ biopolymer proteins are known from teleosts and amphibians (*Schaffeld and Schultess, 2006*), which suggests that thread gene expressing cells in the epidermis may have appeared before the split between jawless and jawed vertebrates, and preceded not only slime glands, but also the hagfishes (*Figure 1B*).

Despite the lack of fossil evidence, anatomical and developmental studies of extant hagfishes suggest that slime glands arose as modifications of the epidermis. An unusual thread-producing cell in hagfish epidermis – epidermal thread cells (ETCs) – is suspected to be homologous to GTCs in the slime glands (*Blackstad, 1963*). Early studies showed that each ETC produces a single thread loosely packed within the cytoplasm (*Schreiner, 1916*), with ultrastructural studies revealing that ETC threads consist of a bundle of filaments that are 8–14 nm in diameter resembling cytoplasmic intermediate

filaments (*Blackstad, 1963*). Moreover, unlike GTCs, ETCs produce a dense mass of granules of unknown function in the distal region of the cell (*Schreiner, 1916*). While ETC threads and granules appear to be secretory products, there is no evidence that they are released via merocrine or apocrine modes (*Blackstad, 1963*).

Therefore, if the threads produced by ETCs are destined for export, a holocrine-like mechanism that depends on cell rupture is more likely. Such mechanism is likely to be passive, because there is no obvious evidence for active release as occurs with the muscle-powered ejection of cells by slime glands. We thus hypothesized that ETCs rupture and release their contents when the skin is damaged, especially during interactions with predators. This is supported from field-captured videos, which show hagfishes are subject to attacks from sharp-toothed predators such as sharks (*Zintzen et al., 2011*). Under natural conditions, hagfishes are likely to sustain frequent damage to their skin from predator bites with their loose and flaccid skin (*Boggett et al., 2017*). In this way, the rupture of epidermal cells may resemble the release of alarm cues during skin damage in lampreys and many jawed fishes (*Pfeiffer and Pletcher, 1964*; *Bals and Wagner, 2012*; *Pandey et al., 2021*).

To explore the functions of ETCs, we examined the abundance of ETCs in hagfish skin and examined the morphology of epidermal threads. We found the epidermal threads are shorter than slime threads but present within epidermis over the entire hagfish body. By experimentally inducing wounds to hagfish skin, we found the damaged epidermis released an adhesive and fibrous slime, which contained threads and granules released from ruptured ETCs. Further, comparative transcriptome analysis from skin and slime glands, together with phylogenetic analysis of α and γ protein sequences, revealed that the thread proteins expressed in hagfish skin are evolutionary sisters to those found in slime glands, with gene duplication and divergence generating a diversity of thread genes uniquely expressed in slime glands. Based on these data, we develop a hypothesis for how slime glands might have arisen from epidermis, as well as the selective pressures that might have driven this transition.

## Results

### Thread and mucous cells cover the entire hagfish body

Within an epidermal thickness of ~95–110 µm, ETCs are generally found in the basal half (~50 µm and deeper) of the epidermis, along with large mucus cells. ETCs and large mucous cells are covered by three to five layers of small mucous cells (*Figure 2A*). Viewing the skin perpendicular to the apical surface (en face view), the outer epidermal surface is covered by densely packed small mucous cells, while the deeper portion contains mainly ETCs and large mucous cells (*Figure 2B*; *Videos 1–2*). To assess the abundance of ETCs over the entire epidermis, we sampled the density of all three epidermal cell types from nine transverse cross-sections from head to tail (*Figure 2—figure supplement 1*). We approximated the area density of each cell type with respect to skin area as $\sigma = \lambda^2$, where $\lambda$ is the linear density sampled from the cross-section of skin.

We found that the proportion of the three cell types varies little across the different regions sampled, with ETCs being the second most abundant. The mean area density of ETCs was ~434 mm$^{-2}$. For an adult hagfish (~45 cm long), we estimate a total of ~$1.2 \times 10^7$ ETCs covering the entire hagfish body. Notably, this total number of ETCs is ~3.9 times greater than the total number of GTCs from all slime glands combined (~$3.1 \times 10^6$, assuming a total of 163 glands; see Materials and methods, Section 'Fibrosity of defensive slime'). In addition, the large mucous cells occurred with density ~92 mm$^{-2}$, which is ~4.7 times lower than that of ETCs. The small mucous cells occurred at a density of $4.3 \times 10^5$ mm$^{-2}$, which is ~1000 times more abundant than the ETCs. (*Figure 2—figure supplement 1*). These abundance data allowed us to approximate the relative proportions of cellular products in epidermal mucus (see below).

### Structure of ETCs

We also examined cross-sections of the skin of *E. stoutii* using laser scanning confocal microscopy and identified three prominent structures within ETCs: (1) a densely packed granule cluster, (2) a thread that is loosely packed along the inner plasma membrane and that interweaves among granules, and (3) a large nucleus located at the basal surface of the granule cluster (*Figure 2C–F*). Such a layout features more unoccupied cytoplasmic space compared to GTCs, in which most cytoplasmic volume

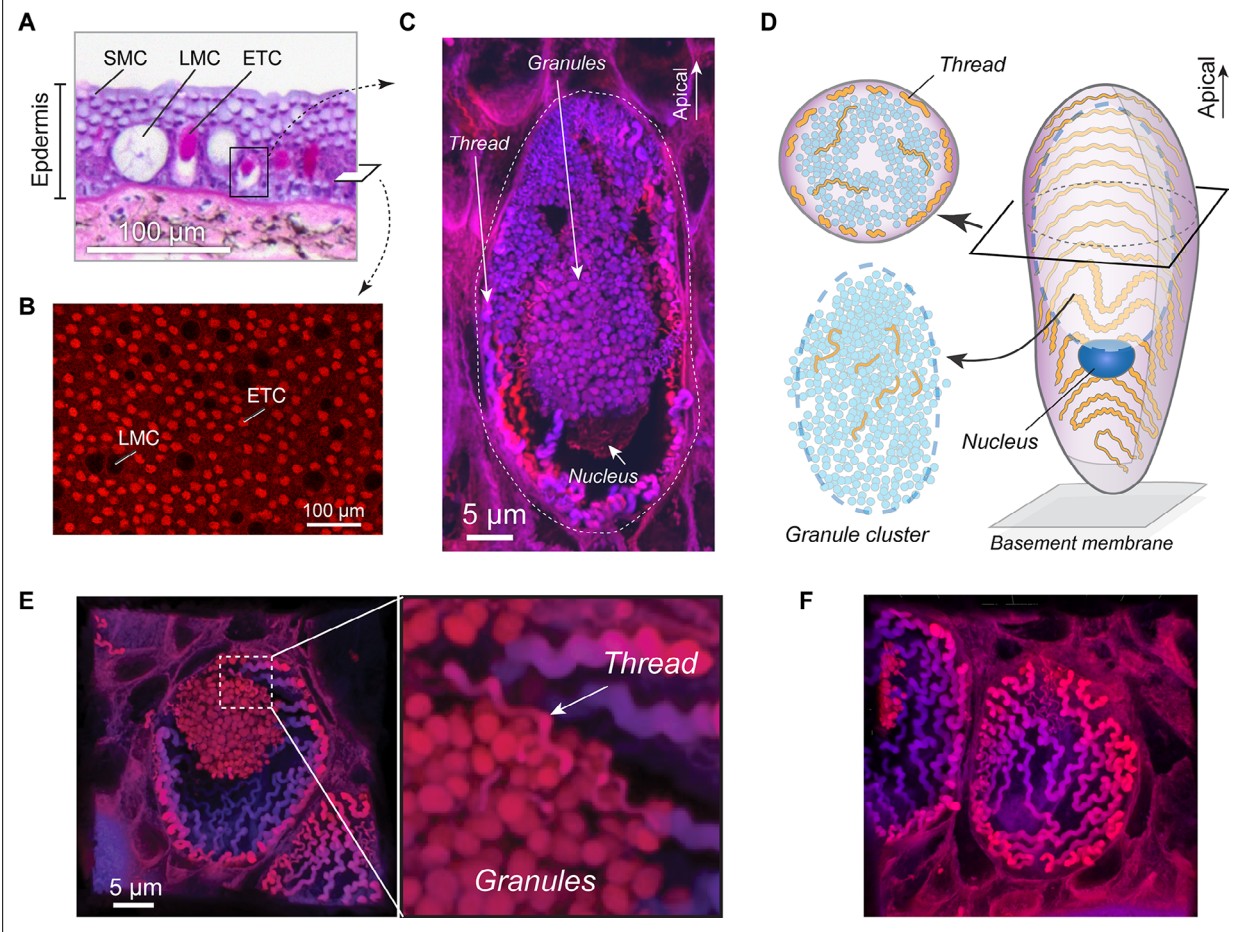

**Figure 2.** Morphology of the hagfish epidermal thread cell. (**A**) Cross-section of dorsal epidermis from a Pacific hagfish (*Eptatretus stoutii*; hematoxylin-eosin-stained; bright-field microscopy). SMC, small mucous cell; LMC, large mucous cell; ETC, epidermal thread cell. (**B**) The basal layer of epidermis containing epidermal thread cells and large mucous cells, as captured in en face view. ETCs are characterized by granules and threads stained with the fluorescent stain eosin; LMCs appear as circular voids. (**C**) Longitudinal cross-section of an ETC, showing a cluster of granules, the nucleus located at the basal region of the granules, and a helical thread located mainly along the inner surface of the plasma membrane. (**D**) Schematic of major cellular components of an ETC. (**E**) Oblique cross-section of an ETC, showing the relative positions of the granule cluster and threads. Enlarged area shows a region where the thread is intimately associated with the granule cluster. (**F**) A close-up of the inner plasma membrane, showing the thread packed in a single layer in a switchback pattern. All images were captured with confocal microscopy unless otherwise noted.

The online version of this article includes the following source data and figure supplement(s) for figure 2:

**Source data 1.** Density of epidermal cells.

**Figure supplement 1.** Abundance of hagfish epidermal cells.

**Figure supplement 2.** Morphology of epidermal thread cell (ETC) at different developmental stages.

is filled by the nucleus and thread skein across different developmental stages (*Figure 2—figure supplement 2*).

The granule cluster dominates the cytoplasm of ETCs and in some cells can span across 80% of the apical-basal axis (*Figure 2C*; *Videos 3–4*). Fluorescence staining with eosin suggests that the ETC granules are composed of protein, but we have no information about the identity of the proteins. Within cross-sections of granule clusters, granule density was $1.05\pm0.50$ $\mu m^{-2}$ (mean ± SD). Although generally round, the granules were not spherical, with an aspect ratio (i.e., the ratio of major to minor axis) of $1.5\pm0.6$ (mean ± SD; $N$=1462 granules).

## Shape and size of epidermal threads

From transmission electron microscopy (TEM) and confocal microscopy, we found three levels of thread structures: (1) At the nanometer scale, TEM images show parallel filaments that are likely

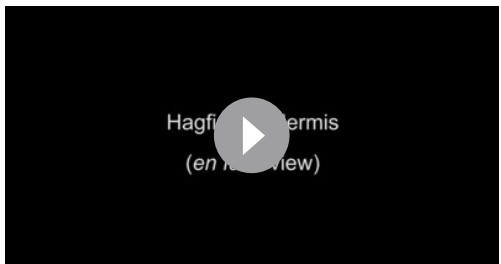

**Video 1.** Z-stack image sequences of eosin-stained hagfish epidermis from confocal laser scanning microscopy with transmitted light, taken in en face view. Note a dense layer of epidermal thread cells (ETCs) and large mucous cells at the basal layer of epidermis. Each ETC is evident with a cluster of granules highlighted in red, while large mucous cells appear as voids.

https://elifesciences.org/articles/81405/figures#video1

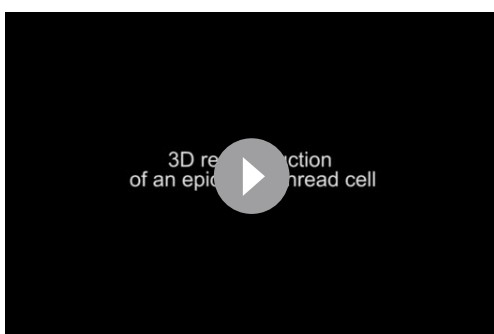

**Video 3.** Z-stack image sequences of hagfish epidermal thread cells (ETCs) based on confocal laser scanning microscopy. Images were taken from eosin-stained epidermis, and the ETC granules are the brightest feature.

https://elifesciences.org/articles/81405/figures#video3

intermediate filaments, which is consistent with previous results (*Blackstad, 1963*; *Figure 3A*). (2) At the micrometer scale, epidermal threads trace regular right-handed helices. (3) At the sub-cellular scale, the helical thread is packed in a single layer in a switchback pattern against the inner plasma membrane surface (*Figure 2F*; *Video 4*). At one of its ends, it is interwoven among granules (*Figure 2E*), a configuration that may contribute to a scaffolding function once ETC contents are released (see below).

All epidermal threads examined were right-handed helices ($N$=25 cells). To understand the helical geometry of threads, we randomly sampled helix sections with centerline lengths of 5–15 μm. We found that the thread diameter ($\phi$) varied between 0.2 and 1.0 μm (0.52±0.18 μm; mean ± SD), which is ~25% of the mean diameter of slime threads (~2 μm). The helical pitch angle ($\theta$) varied between 47.6° and 81.8° (63.5°±5.6°) and was relatively consistent across the full thread diameter range for a given segment of thread. Similarly, the helical diameter ($D$) varied between 0.07 and 0.78 μm (0.35±0.10 μm), with a slight reduction with increasing $\phi$ (*Figure 3B–C*). The pitch angle allowed us to calculate how much the threads can increase in length if the helix is pulled taut. The

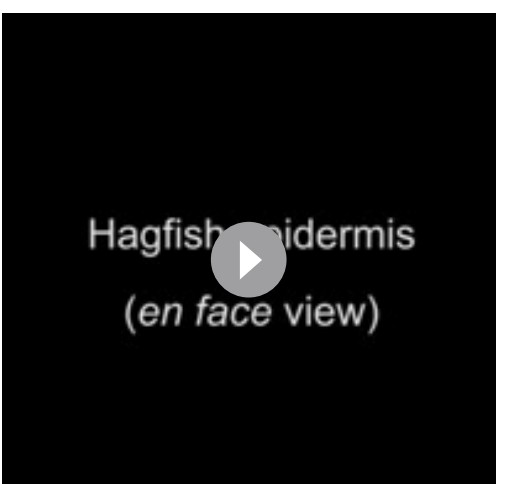

**Video 2.** Z-stack image sequences of hagfish epidermis from confocal laser scanning microscopy, taken in en face view. Note the outermost epidermis is covered by a layer of small mucous cells, while epidermal thread cells (ETCs) are found at the basal layer.

https://elifesciences.org/articles/81405/figures#video2

**Video 4.** Three-dimensional reconstructions of epidermal thread cells (ETCs) based on confocal laser scanning microscopy, showing granule cluster and helical-shaped threads packed along the plasma membrane.

https://elifesciences.org/articles/81405/figures#video4

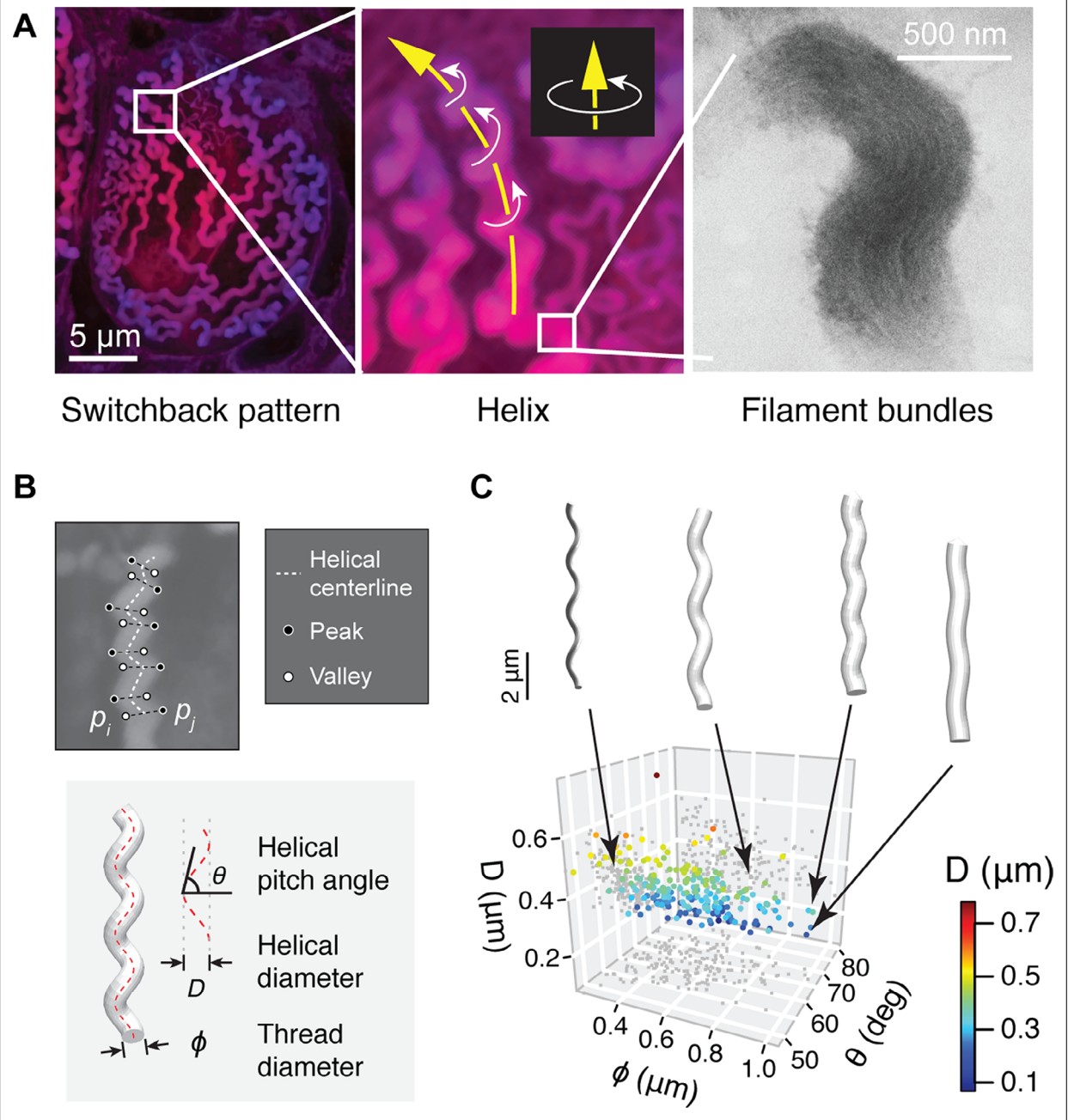

**Figure 3.** Geometry of hagfish epidermal threads. (**A**) Three levels of epidermal thread structure. (Left-middle) At the micro-scale, the thread traces a right-handed helix, the centerline of which is arranged in a switchback pattern on the inner surface of the cell membrane. Yellow arrow denotes the direction of increase; white arrows denote direction of helical rotation. (Right) At the nano-scale, a thread consists of a dense bundle of intermediate filament proteins, shown here in transmission electron microscopy (TEM) (see also *Figure 3—figure supplement 1*). (**B**) The peaks and valleys of the projected thread sections were used as landmarks for morphometric analysis. Blue dots, peaks; white dots, valleys; white dashed line, centerline. $\phi$, thread diameter; $\theta$, helical pitch angle; $D$, helical diameter. (**C**) Variations in thread geometry with respect to a morpho-space defined by thread diameter $\phi$, helical pitch angle $\theta$, and helical diameter $D$. With increasing pitch angle $\theta$, thread diameter $\phi$ increases ($p<0.05$; linear regression model) and helical diameter $D$ decreases ($p<0.001$), illustrated with idealized threads.

The online version of this article includes the following source data and figure supplement(s) for figure 3:

**Source data 1.** Geometry of epidermal threads sampled using laser confocal microscopy.

**Figure supplement 1.** Morphology of epidermal threads.

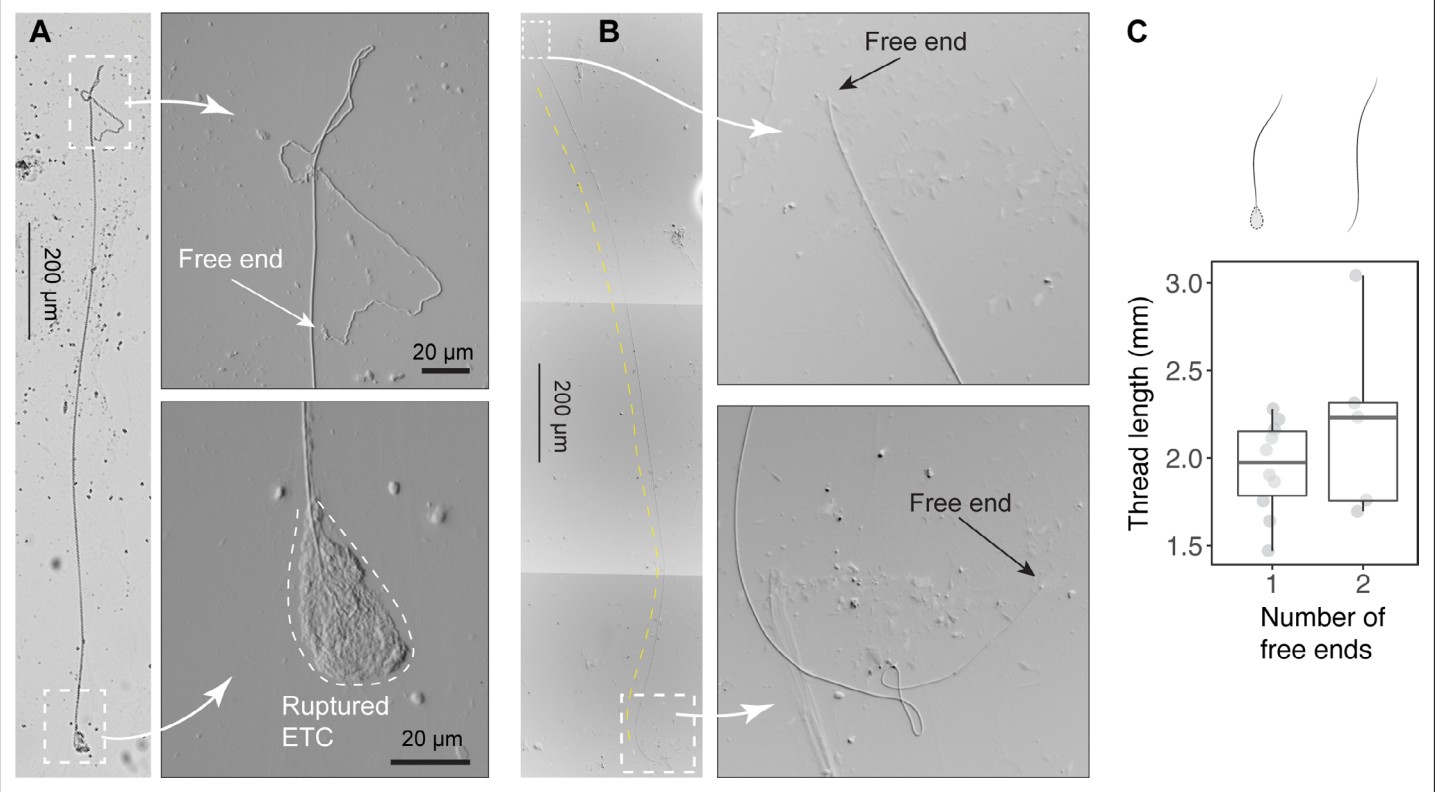

**Figure 4.** Size of released epidermal threads. (**A**) A partially released thread (~2 mm long) from a ruptured epidermal thread cell (ETC), as viewed under light microscopy (see also *Figure 4—figure supplement 1*). (**B**) A thread with two free ends. (**C**) Boxplots of thread length ($L_T$) measurements based on threads with one free end (N=10) and two free ends (N=5). Values are length from individual threads. For threads with one free end, $L_T$ = 1.95 ± 0.27 mm (mean ± SD); for threads with two free ends, $L_T$ = 2.2 ± 0.54 mm, with this mean value used in scaling models.

The online version of this article includes the following source data and figure supplement(s) for figure 4:

**Source data 1.** Length of epidermal threads sampled using transmitted light microscopy.

**Figure supplement 1.** Release of epidermal threads.

**Figure supplement 2.** Release of epidermal threads.

extension can be characterized by an extension ratio $R_{Ext} = 1 - \sin\theta$, which averaged 10.1% over the range of pitch angles described above.

## Epidermal threads versus slime threads

Due to the complex shape of threads, their long aspect ratios, and the difficulty of tracing threads among granules, we were not able to reconstruct the morphology of an entire thread using confocal microscopy. We were able, however, to measure the full length of threads we collected by scraping hagfish skin with a cover glass. Thread length $L_T$ from these measurements was 2.2±0.54 mm (mean ± SD; *Figure 4*). These isolated threads were generally straight and showed little evidence of the helical morphology seen in intact ETCs. Incorporating the helical pitch angle $\theta$ above, we can approximate the total length of the helical centerline as $L_T' = L_T \sin\theta$ =1.97 ± 0.10 mm, which is ~40 times longer than the cell's major axis (~50 µm). Overall, the epidermal threads are ~90 times shorter and ~4 times thinner than slime threads, but are still one of the largest intracellular fibers known (*Figure 5*). Some epidermal threads appeared to cleave into multiple sub-threads after being stretched, suggesting loose inter-filament binding (*Figure 5B–C*).

Assuming threads are cylindrical and ETCs are ellipsoidal, the volume fraction occupied by threads within the cytoplasmic space can be approximated. Using the ranges of thread radius ($r_T = 0.5\phi$), thread length $L_T$, and mean cell dimensions (major axis $r_a$ ~27 µm; minor axis $r_b$ ~23 µm; see *Figure 2— figure supplement 2*), we found the epidermal threads only occupy 1.4–5.8% of the cytoplasmic space, which is much lower than the GTCs, where thread skeins may occupy >95% of the cytoplasmic

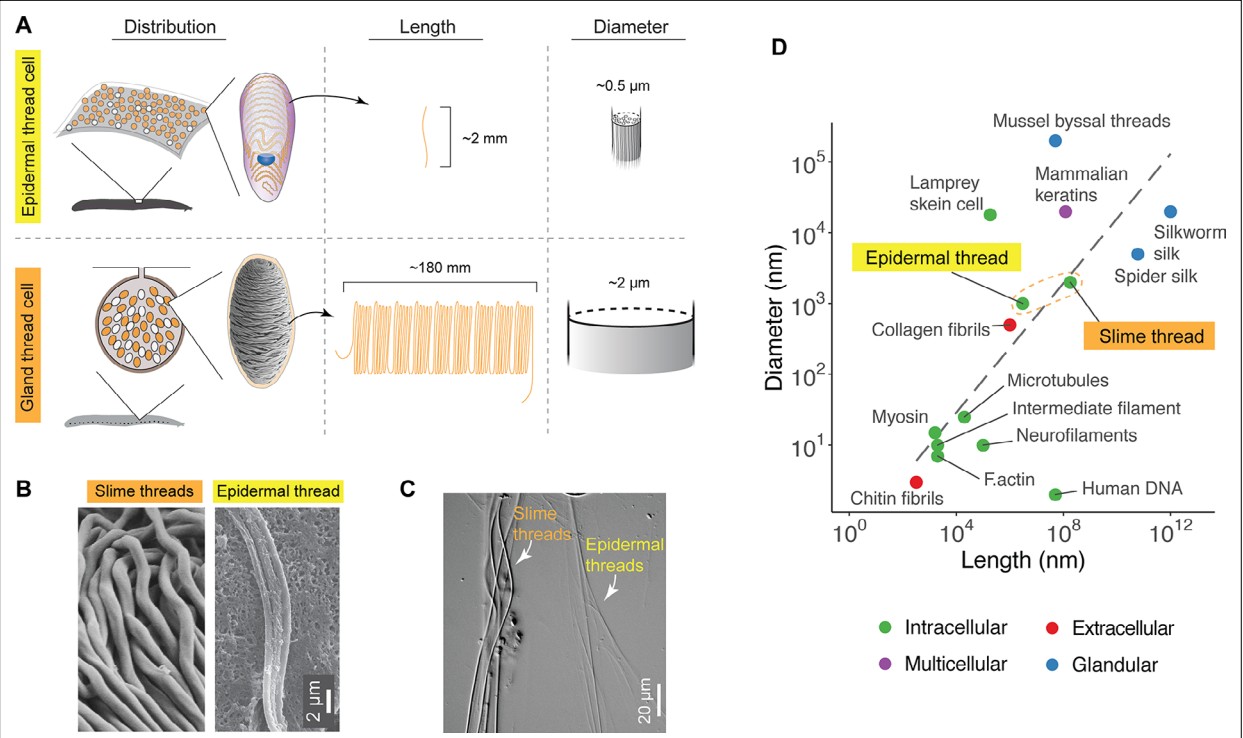

**Figure 5.** Comparison of distribution, size, and shape between epidermal and slime threads. (**A**) A schematic comparison of the distribution of epidermal and gland thread cells, as well as the mean lengths and diameters of corresponding thread products. Data based on Pacific hagfish (*Eptatretus stoutii*). (**B**) A comparison of slime and epidermal threads viewed with scanning electron microscopy (SEM). Note the epidermal thread has appeared to cleave into multiple sub-threads after being stretched (see also *Figure 3—figure supplement 1*). (**C**) Two types of threads collected from the same hagfish (viewed with differential interference contrast microscopy), highlighting their difference in diameter. (**D**) A size comparison of epidermal and slime threads in Pacific hagfish (*E. stoutii*) with other biofibers. Trend line represents a linear regression model based on all data points excluding human DNA. Colors denote different fiber production mechanisms (see *Zeng et al., 2021*).

space (*Downing et al., 1981*; *Zeng et al., 2021*). To assess the thread storing capacity of the skin, we combined the stored thread length and area density of ETCs to calculate the area density of threads: $\sigma_T = \sigma_{ETC} L_T$, which yields a total of ~96 cm of threads per square millimeter of skin.

## Damaged skin produces a fibrous slime

Dragging a sharp pin across a hagfish's skin resulted in the formation and accumulation of a thick epidermal slime (translational speed ~17 cm/s; mean vertical force 0.06 N, pressure ~2 MPa, assuming a contact area of 0.03 mm²; *Figure 6A and B*; *Video 5*). Examining the path of the pin on the skin with scanning electron microscopy (SEM) revealed evidence that scraping caused rupture of ETCs and release of granules and threads. In relatively shallow wounds, where only the apical portions of ETCs were removed, the granule-thread complex was typically found anchored with the basal portion of threads to the inner surface of the cell's plasma membrane (*Figure 6C and D*).

Epidermal slime appeared as a white material that adhered to the scraping object, exhibiting properties distinct from the defensive slime (*Videos 5 and 6*). Examination of the slime with light microscopy and SEM confirmed the presence of granules and threads, along with threads aligned with the scraping direction (*Figure 6E*; *Figure 6—figure supplements 1–3*). Released granule-thread complexes were observed on the edge of coverslips used for scraping or on the skin surface after scraping, and often were seen with a single thread trailing from a granule cluster (*Figure 4—figure supplement 2*). Although we saw no direct evidence of cell products from large mucous cells, given their position in the same basal layer of the epidermis, it is likely that large mucous cells rupture under the same conditions that cause ETC rupture and contribute to the mucus components of epidermal slime.

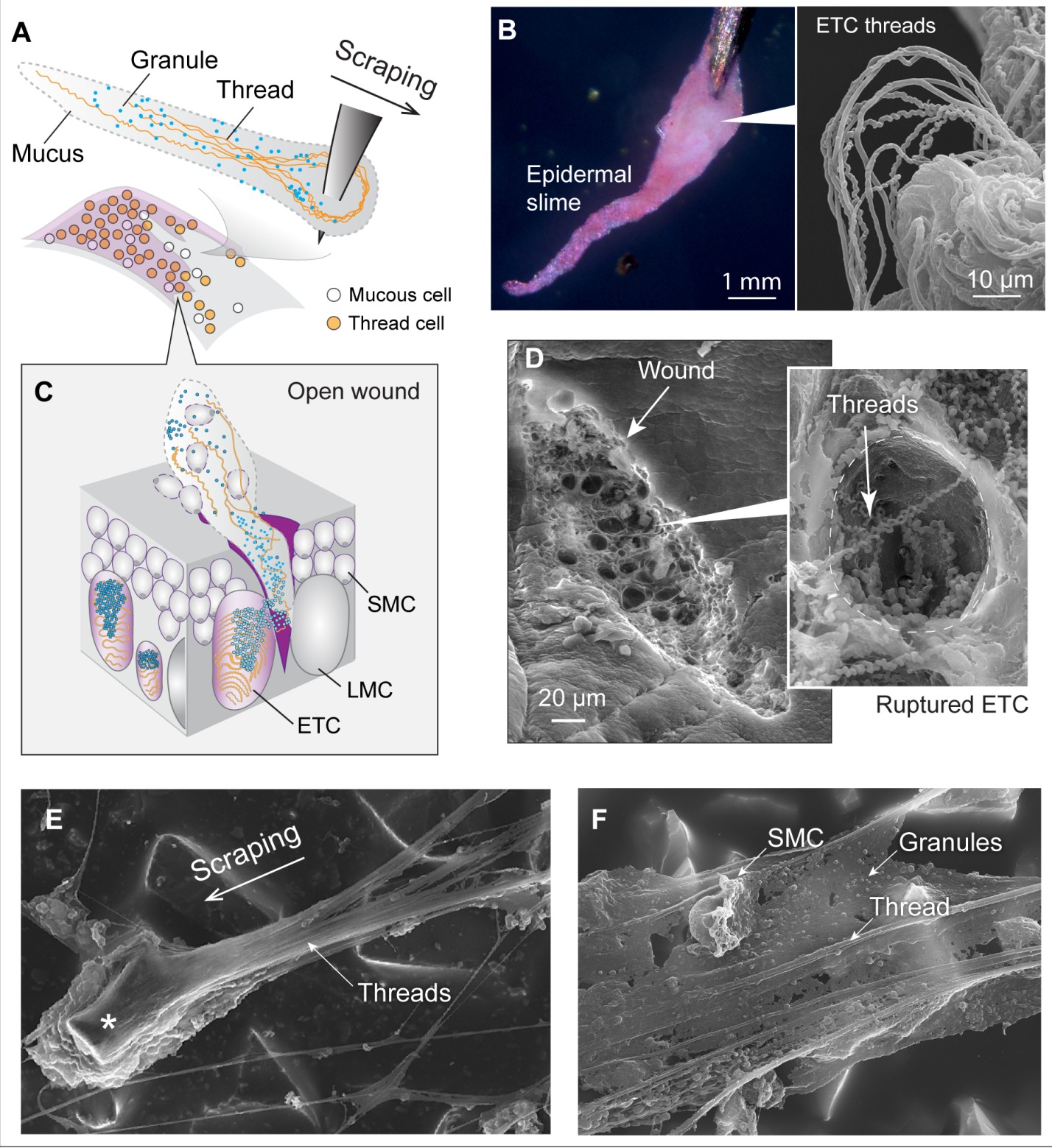

**Figure 6.** Formation and structure of epidermal slime produced by wounded skin. (**A**) Schematic of epidermal slime formation when epidermis is wounded, with threads and granules from ruptured epidermal thread cells (ETCs) mixing with mucus from ruptured large mucous cells (LMCs). (**B**) Epidermal slime on pin tip, stained with eosin to show threads. (Right) Scanning electron microscopy (SEM) image of epidermal slime on pin tip, with enlarged areas showing stretched and unstretched threads. (**C**) A schematic of the slime formation by mixing of cellular contents from an open wound on epidermis. SMC, small mucous cell. (**D**) SEM images of a shallow abrasion wound, with insets showing damaged ETCs with partially released

*Figure 6 continued on next page*

*Figure 6 continued*

threads and granules. (**E**) Epidermal slime collected on sandpaper. Note the slime accumulated at the leading edge of the sand grain (asterisk) and the elongated slime at the trailing edge. (**F**) Thin film of epidermal slime collected by scraping with sandpaper, showing the scaffolding of mucus by threads, and the alignment of threads with the scraping direction. See more details in *Figure 6—figure supplements 1–3*.

The online version of this article includes the following figure supplement(s) for figure 6:

**Figure supplement 1.** Formation of epidermal slime.

**Figure supplement 2.** Structure of epidermal slime.

**Figure supplement 3.** Structure of epidermal slime.

## Epidermal slime versus defensive slime

Scraping with the edge of a cover glass over 18 cm² of skin that had been blotted dry led to about 2–10 mg (5.2±2.4 mg; mean ± SD) of slime adhered to the coverslip, which is equivalent to a productivity of ~0.3 mg/cm². The relative water content of epidermal slime sampled from skin immersed in seawater ranged from 92% to 96% (93.9% ± 1.2%; mean ± SD) and from 70% to 90% (74.7% ± 6.8%) for samples collected from skin that was blotted dry (*Figure 7A*).

Both epidermal slime and defensive slime are structurally heterogeneous, containing long threads and mucus. Here, we use the ratio between the total thread length and the total slime volume to characterize the level of 'fibrosity' of the two types of slime. The fibrosity index of epidermal slime was calculated as:

$$r_F = \frac{L_T}{V_S} \qquad (1)$$

where $L_T$ is the total length of thread and $V_S$ is the volume of slime. Specifically, $L_T$ was calculated as the product between the mean length of a single thread $L_T'$ and the number of ETCs: $L_T = L_T' N_{ETC}$ .

Considering an ideal situation without swelling with seawater, the volume of slime should be equal to the total volume of ruptured epidermal cells. With a unit skin area $A$ and the mean thickness of epidermis $D_{epi}$ = 100 μm, we have $V_{S(Unswollen)} = A D_{epi}$ and $N_{ETC} = \sigma_{ETC} A$. Thus, *Equation 1* can be expressed as:

$$r_{F(Unswollen)} = \frac{L_T' \sigma_{ETC}}{D_{epi}} \qquad (2)$$

Incorporating the single thread length ($L_T'$ = 2.2 mm) and the area density of ETCs ($\sigma_{ETC}$ = 434 mm⁻²), we found $r_{F(Unswollen)}$ ≈ 9600 mm/mm³ for epidermal slime without swelling with seawater. Next, acknowledging that swollen slime has ~19% more water than unswollen slime and assuming the density of unswollen slime is close to that of seawater, we derived $V_{S(Swollen)} = 1.19 V_{S(Unswollen)}$ and approximated $r_{F(Swollen)}$ ≈ 8024 mm/mm³ for swollen epidermal slime, which is ~686 times higher than that of the defensive slime (~12 mm/mm³; based on *Schorno et al., 2018*; see Materials and methods, Section 'Fibrosity of defensive slime'). Together, these results show that epidermal slime is less dilute and much more fibrous than defensive slime (*Figure 7B and C*).

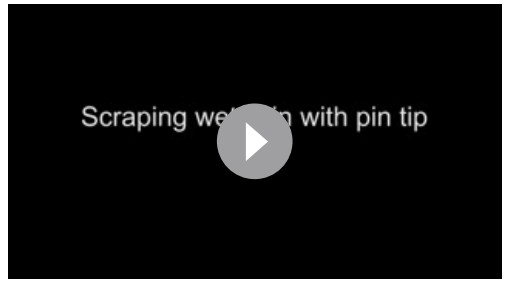

**Video 5.** Experimentally induced formation of epidermal slime, demonstrated by scraping wet and blot-dried hagfish skin with a sharp pin head. Scraping with a blunt pinhead did not lead to slime formation.
https://elifesciences.org/articles/81405/figures#video5

**Video 6.** Different from condensed, adhesive hagfish epidermal slime, the defensive slime is highly diluted and not sticky.
https://elifesciences.org/articles/81405/figures#video6

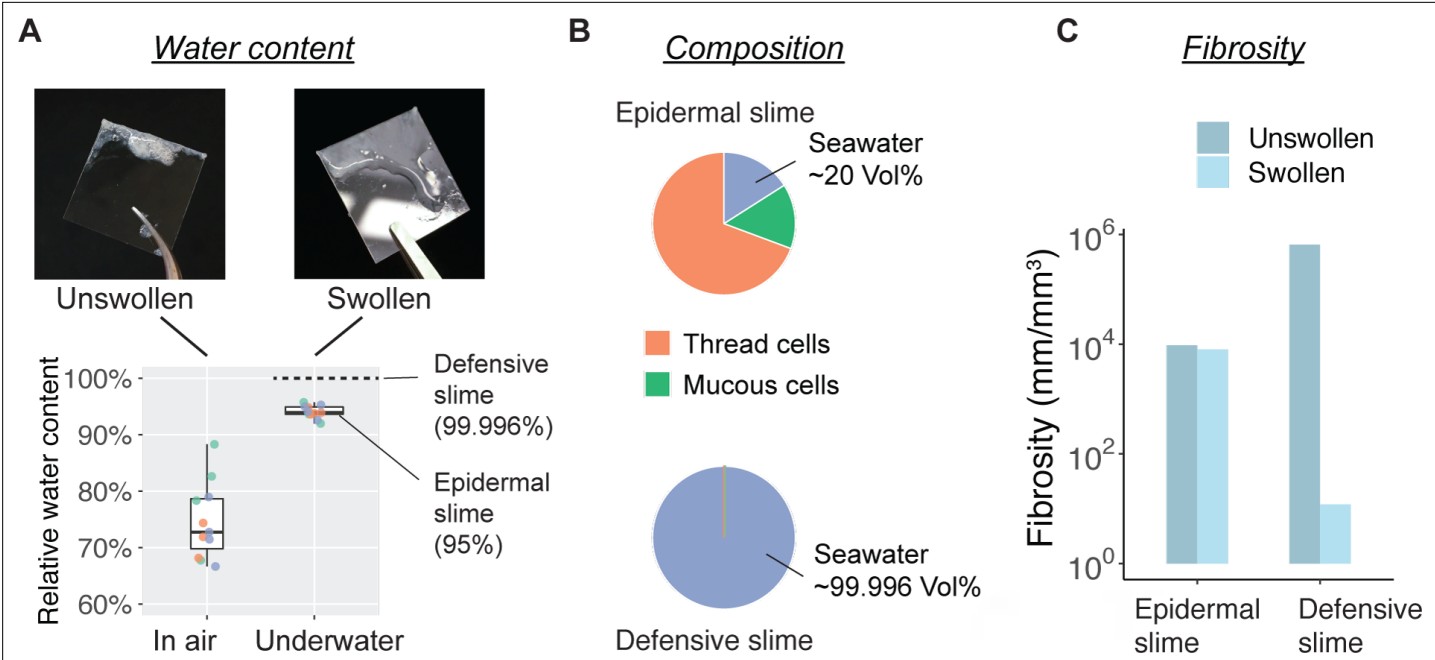

**Figure 7.** Water content and fibrosity of hagfish epidermal slime. (**A**) The relative water content of epidermal slime collected by scraping a glass coverslip over blotted skin (unswollen) and underwater (swollen). Dots represent individual samples; colors represent different animals (*N*=3 for each group; see Materials and methods). (**B–C**) A comparison of slime composition (in relative volumes) and fibrosity between epidermal and defensive slimes. Note the high water content and low fibrosity of defensive slime produced with turbulent mixing after active ejection. See *Figure 7—figure supplement 1f* or more information on defensive slime.

The online version of this article includes the following source data and figure supplement(s) for figure 7:

**Source data 1.** Water content of epidermal slime sampled from hagfish skin (blot-dried in air versus underwater).

**Figure supplement 1.** Morphometrics of hagfish defensive slime.

## Slime thread genes are derived from duplications of epidermal thread genes

We examined the transcriptomes of skin and slime glands. Two types of thread proteins, α and γ, were previously identified (*Koch et al., 1994*; *Koch et al., 1995*) and threads produced from these genes were hypothesized to comprise the fibrous slime of hagfish. We characterized α and γ thread transcripts from replicate RNAseq datasets from skin and slime gland tissues of *E. stoutii* and a close relative, *E. goslinei*.

Transcriptomic and phylogenetic analyses of hagfish α thread transcripts from both species identified either a single (*E. goslinei*) or a low diversity of α thread biopolymer transcripts (*E. stoutii*) that are highly expressed in the epidermis and likely comprise a portion of the epidermal thread biopolymers (*Figure 8*). We also uncovered a monophyletic diversity of highly expressed slime gland-specific α transcripts for both species. Analyses of γ thread biopolymer transcripts show a strikingly similar pattern for both species. Only a single skin-expressed γ thread transcript was identified in *E. stoutii* and no γ thread transcript was recovered from *E. goslinei* skin transcriptomes, however, as with α thread transcripts, both species display a large monophyletic diversity of slime-gland expressed γ thread transcripts. These data suggest that rampant, hagfish-specific, gene duplications of GTC-specific α and γ thread genes played a role in the evolution of defensive slime in hagfishes.

## Discussion

Our results demonstrate that epidermal threads are released through cell rupture during skin damage, together with mucus, forming a fibrous epidermal slime. With the ETC granules possibly serving anti-predator, antimicrobial or alarm functions (see below), the epidermal slime can be produced during interactions with predators and likely represents an incipient form of the defensive slime. Also, gene

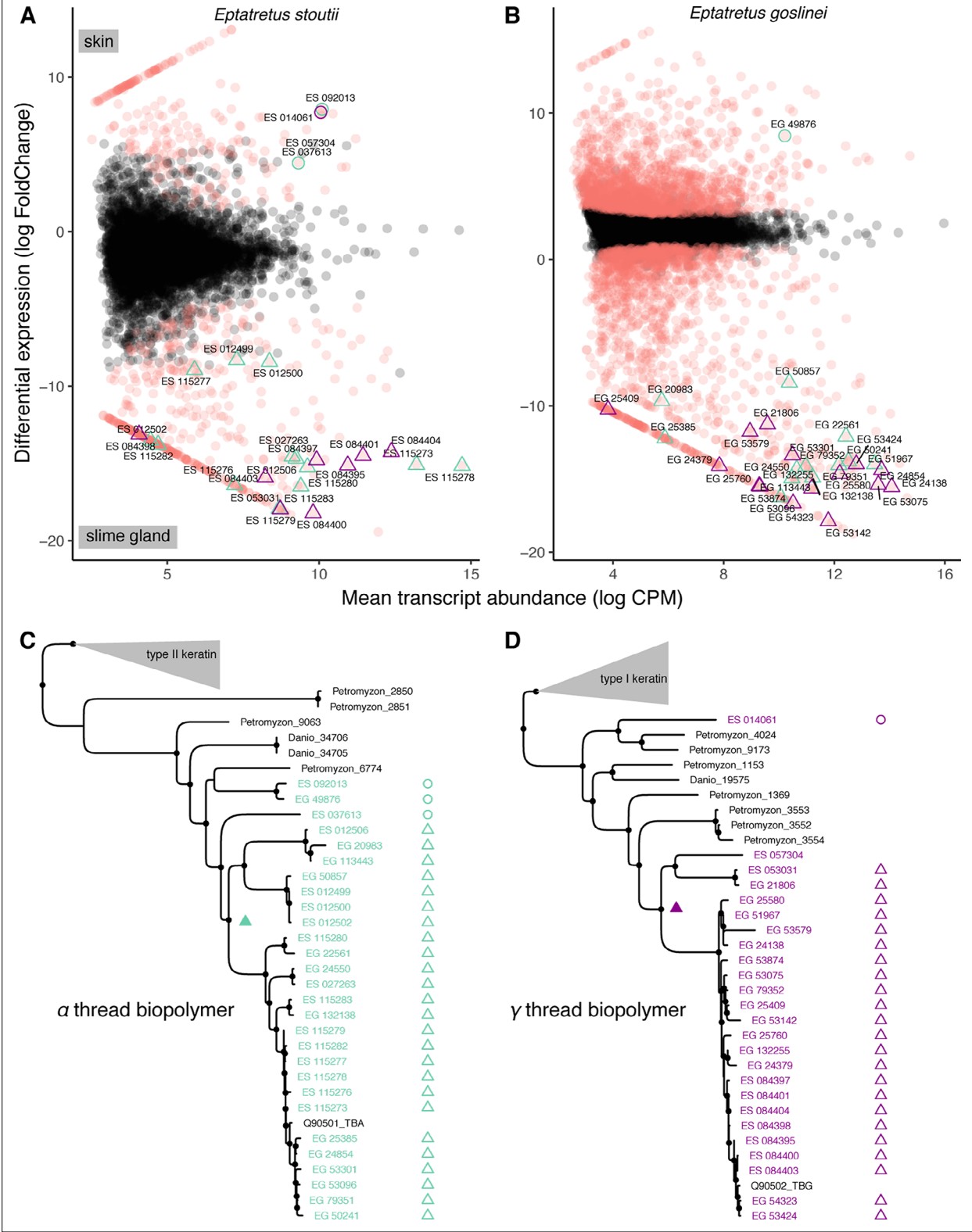

**Figure 8.** Molecular analyses suggest an epidermal origin of hagfish defensive slime threads. (**A and B**) Differentially expressed transcripts (red) from skin versus slime gland RNAseq datasets (3× replicates each, from single specimens of *Eptatretus goslinei*; FDR <0.001). In both species, a low diversity of α and γ thread biopolymer genes are expressed in skin, while a large diversity of both α and γ thread biopolymer genes are expressed in slime gland. (**C and D**) Comparative phylogenomic analyses of α and γ thread gene trees (maximum likelihood) reveal slime gland- and hagfish-specific expansions of both α and γ intermediate filament genes. The presence of well-characterized, skin-specific α and γ thread orthologs from both lamprey (*Petromyzon*)

*Figure 8 continued on next page*

*Figure 8 continued*

and teleost (*Danio*) indicates that independent gene duplications of skin-expressed α and γ loci gave rise to a radiation of slime gland-specific transcripts. Open circles indicate skin expression; open triangles indicate slime gland expression; filled triangles indicate most parsimonious prediction of transition from ancestral skin expression to slime gland expression; small circles at nodes indicate greater than 80% support from either ultrafast bootstrap approximation or approximate likelihood ratio tests (*Nguyen et al., 2015*). Trees shown here are pruned from larger phylogenetic analyses. See *Figure 8—figure supplements 1–2* and Materials and methods for details.

The online version of this article includes the following figure supplement(s) for figure 8:

**Figure supplement 1.** Phylogenetic analysis of vertebrate thread biopolymer alpha genes from selected taxa, rooted with its closest sister clade, which is comprised of type II keratins.

**Figure supplement 2.** Phylogenetic analysis of vertebrate thread biopolymer gamma genes from selected taxa, rooted with its closest sister clade, which is comprised of type I keratins.

expression data and phylogenetic analysis support an epidermal origin of slime glands. Below, we discuss the structure and function of epidermal threads and propose a model to explain the origins of hagfish slime glands and defensive slime.

## Possible functions of ETC granules

While the thread is the most conspicuous part of an ETC when viewed with conventional histology and microscopy (hence their name), our measurements based on high-resolution confocal microscopy show that the granules take up far more volume in the cell than the thread. The production of granules in large numbers is typical of secretory cells (*Bowen, 1929*) and further suggests a secretory function served by ETCs (*Blackstad, 1963*; *Spitzer and Koch, 1998*). The lack of any obvious secretory mechanism for ETCs and the results of our skin wounding experiments lead us to the conclusion that granules are primarily released when the epidermis is damaged and ETCs are ruptured. In light of these results, we consider three possible functions for the granules: (1) They may contain distasteful compounds that help deter predators when hagfishes are bitten. A similar defensive function has been suggested for the epidermal granule cells of lampreys (*Pfeiffer and Pletcher, 1964*) and this would be a reasonable adaptation for hagfishes, whose scavenging lifestyle involves frequent bites from predators (*Zintzen et al., 2011*; *Boggett et al., 2017*). (2) The granules may contain antimicrobial compounds that help prevent infection after the skin is damaged (e.g., 'myxinidin'; *Subramanian et al., 2009*). This too would be sensible for an animal that is frequently bitten. (3) The granules may contain an alarm pheromone that alerts other hagfishes to the presence of an attacking predator, a mechanism that has been widely reported in lampreys and fishes (*Bals and Wagner, 2012*; *Imre et al., 2014*; *Pandey et al., 2021*). The identity of the proteins that make up ETC granules is unknown, but future work to identify and characterize these proteins will undoubtedly shed additional light on their function.

## Structure and function of epidermal threads

Like slime threads, epidermal threads appeared to be mechanically robust, with no evidence of threads breaking even when they were sheared under a cover glass. The lack of helical twists in the elongated threads suggested that the threads are capable of plastic deformation, a property that has also been observed in slime threads and individual intermediate filaments (*Fudge et al., 2003*; *Kreplak et al., 2005*; *Forsting et al., 2019*). Notably, the appearance of loose subfilament structure in some epidermal threads (*Figure 5B*; *Figure 3—figure supplement 1*) has not been observed in slime threads. This suggests that epidermal threads may simply be a bundle of individual intermediate filaments. In contrast, intermediate filaments in slime threads undergo a phase transition in which filaments condense with their neighbors to form a single, electron-dense thread (*Winegard et al., 2014*; *Terakado et al., 1975*; *Downing et al., 1984*).

The production of a macroscopic thread that is released after cell rupture suggests an evolutionary affinity between ETCs and GTCs and provides support for an epidermal origin of slime glands. If GTCs were derived from an ancestral ETC-like cells, selection for greater thread length and strength (and therefore diameter; *Figure 3B*) was likely responsible for the evolution of a tightly packed thread skein and accordingly the loss of granules in GTCs (see *Zeng et al., 2021*).

Our results show that epidermal threads associate with mucus to form a fibrous epidermal slime, which may be the evolutionary precursor of defensive slime (see below). While the length of individual

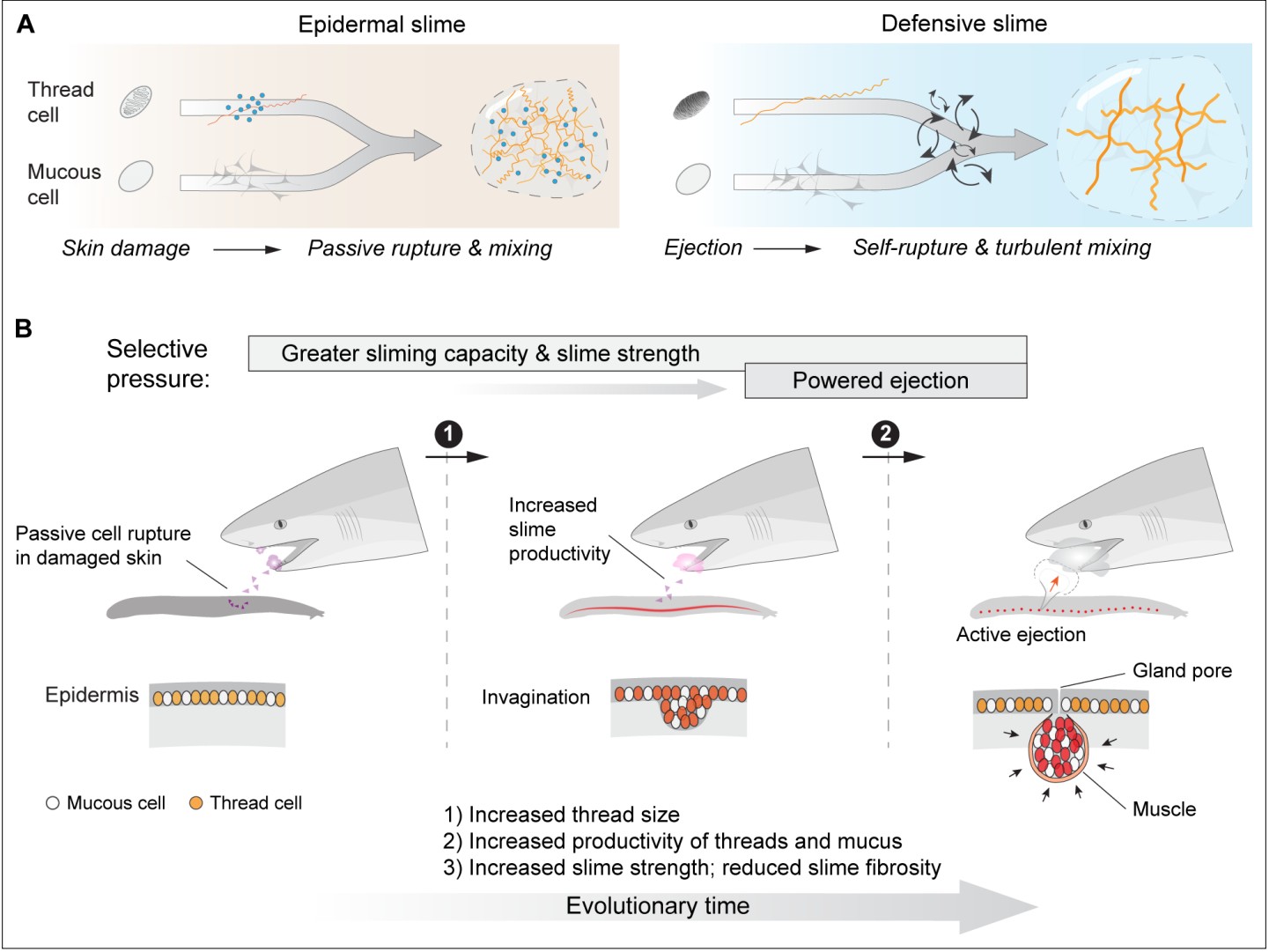

**Figure 9.** An epidermal origin of hagfish slime. (**A**) A comparison of slime formation mechanism between epidermal and defensive slimes, highlighting their similarity in basic structural components and differences in mixing mechanism. Note a transition from passive slime formation to active ejection, as well as a transition in slime composition. (**B**) Schematic of two critical transitions in the evolution of hagfish slime glands. Specifically, selection for greater slime capacity likely drove an increase in the concentration of thread cells and mucous cells in epidermis and later in slime glands, while selection for active ejection likely was responsible for the acquisition of gland musculature and an enlarged gland cavity with a narrow pore (see Discussion). Bottom row highlights the invagination of epidermis (middle) as a possible intermediate state between the ancestral form (left) and muscularized slime glands seen in modern hagfishes (right).

epidermal threads is small compared to that of slime threads, the large number of ETCs in the epidermis represents a significant reserve of thread length. For example, the total length of epidermal threads produced by ~1.1% of the skin area of an adult hagfish (~3.15 cm²) equals the total length of slime threads ejected from a single slime gland (~2736 m) (see *Figure 7—figure supplement 1*). These numbers demonstrate that a version of the most complex part of hagfish slime – the threads – was likely being produced in large quantities in the skin long before the slime glands appeared.

## Mechanism of epidermal slime formation

Our mechanical abrasion experiments demonstrated the formation of a thick epidermal slime, which shared similar structural components with the defensive slime (*Figure 9A*). Epidermal threads not only appeared to hold the slime together, but they were also readily caught on and adhered to the hard structures we used to damage hagfish skin (i.e., pin, coverslip and sandpaper; *Figure 6*; *Figure 6— figure supplement 3*). Under natural conditions, this property of the epidermal slime may allow it

to adhere to a predator's teeth after it has bitten a hagfish. Once bound to the predator's teeth, the epidermal slime may deliver distasteful compounds to discourage further bites. It is also possible that the slime remains adhered to the hagfish's skin after an attack (see *Figure 6—figure supplement 1*), which would be consistent with an antimicrobial function of ETC granules, with compounds in the granules inhibiting bacterial growth at the wound site. Both the distasteful and antimicrobial hypotheses of epidermal slime function should be tested with further experiments.

## Thread proteins in skin and slime glands

The simplest interpretation of our transcriptomic and phylogenetic analyses is that slime threads evolved from epidermal threads, with duplication and diversification of skin-specific α and γ genes and the novel expression of thread gene duplicates in slime glands (*Figure 8*). Our analyses indicate that skin-expressed α and γ transcripts demonstrate low phylogenetic diversity, but slime gland-expressed α and γ transcripts show high phylogenetic diversity for both species. We hypothesize that some of the exotic biophysical properties of defensive slime may be based on this elevated transcriptomic diversity of α and γ transcripts that are uniquely expressed in slime glands.

Two lines of evidence suggest that independent radiations of GTC-specific α and γ thread genes are each derived from ancestrally skin-expressed loci. First, the presence of skin-specific α and γ thread orthologs from both lamprey and teleosts (which lack slime glands) indicates that skin is the ancestral expression domain for both gene families and that gene duplications of skin-expressed α and γ loci gave rise to a lineage-restricted radiation of slime gland-specific α and γ transcripts in the hagfishes (*Figure 8*). Second, the α and γ thread gene families are themselves sisters to the keratin type II cytoskeletal, and keratin type I cytoskeletal gene families respectively, which are both well characterized in skin. Together, these findings support the origin of slime gland GTCs from skin ETCs.

Orthologs of α and γ biopolymer genes are present in teleosts and amphibians (*Schaffeld and Schultess, 2006*), but the epidermis of these clades is not known to possesses thread bearing cells. Therefore, it is likely that the origin of epidermal expression of α and γ biopolymer genes dates to the last common ancestor of vertebrates while thread bearing ETCs is a cyclostome innovation that subsequently gave rise to slime gland GTCs (*Figure 1B*).

## Implications for the origin of hagfish slime

The morphological, functional, and genetic evidence laid out above are all consistent with an epidermal origin of hagfish slime glands. The GTCs most likely arose via modifications of the ancestors of ETCs. Using the extant form of ETCs as a reference, the transition to GTCs likely comprised an increase in cell size (i.e., GTCs are ~40 times larger than ETCs in volume), an increase in thread diameter and length and the evolution of a highly packed thread skein. How and when thread genes occurred in relation to these transitions remains an open question. While the origin of gland mucous cells was not investigated in this study, the most likely explanation is that they arose from modifications of epidermal large mucous cells, which are also large cells containing numerous mucous vesicles (*Figure 2A*). Despite the two major cell types found in slime glands, there is not a third cell type corresponding with the small mucous cells, which constitutively secret mucus as a protective barrier at the outer skin surface (*Patzner et al., 1982*). During the evolution of slime glands through possible invagination of the epidermis (see below), cells specialized for slow release of mucus had little purpose and were likely excluded in favor of larger proportions of thread cells and large mucous cells.

In addition to explaining changes at the cellular level, a satisfying model of slime gland evolution from epidermis also needs to account for larger scale morphological changes, especially changes in tissue dimensions and the association with striated muscle. Below, and in *Figure 9B*, we describe a possible evolutionary scenario that can be broken down into three major phases.

1. The ancestral form might have resembled the epidermis of modern hagfishes, with thread and mucous cells that when ruptured from external forces (e.g., abrasion, laceration, and puncture) could release a thick and fibrous slime that deterred predators and/or served to inhibit microbial growth. Next, selection for a greater capacity to produce this protective slime promoted the local expansion of the epidermis.

2. Swollen or invaginated skin with enhanced sliming capacity. This intermediate stage may have resembled an exocrine gland with a cavity for temporary storage of thread- and mucus-producing cells or products. Expansion or invagination of the epidermis allowed for increased production and storage of secretory cells, but also created a new challenge of how to more

effectively deploy the slime, such as an increased rate of release and at a specific location along the body.

3. Enlarged, muscularized slime glands with narrow ducts and pores. Selection for rapid, controlled release of a large number of thread and mucous cells likely led to an increased association with striated muscle fibers, which eventually became the dense basket of muscle fibers that envelops the slime gland capsule (i.e., the *musculus decussatus*). Other innovations in this phase may have included the appearance of gland interstitial cells, which may have allowed for delivery of nutrients to the gland interior (see *Fudge et al., 2015*).

While our evolutionary model offers a preliminary framework for the transition from epidermis to slime glands, there remain other open questions regarding the morphology and behavior of slime glands – for example, what determined the number and density of slime glands and why they are aligned as single rows along the lateral side of body. Future work will also explore the morphological and genetic changes associated with the large mucous to gland mucous cell transition as well as the origin of the gland musculature.

Lastly, while an epidermal origin of hagfish slime glands is consistent with the morphological and molecular data presented here, it is not the only possible explanation. Another possibility is that hagfish defensive slime glands arose from cloacal glands, which are found in the dorsal wall of the cloaca and contain both thread and mucous cells (*Tsuneki et al., 1985*). The cloacal glands have only been observed in ripe individuals and are believed to be involved in reproduction. It is possible that cloacal thread cells are the most proximal ancestor to GTCs, arising first for a role in reproduction and only later being co-opted for defense. We did not have access to ripe hagfishes and therefore were not able to examine the morphology and gene expression of cloacal glands. While we haven't been able to rule out the possibility that slime glands originated first in the cloaca, our genetic data and analysis are consistent with an epidermal origin for slime thread genes, and by extension, an epidermal origin of slime glands. Identifying other intermediate forms between epidermis and slime glands, possibly by studying the development and gene expression in slime gland tissue, or finding new intermediate fossils, may help to further clarify the evolution pathway from hagfish epidermis to slime glands.

# Materials and methods

## Key resources table

| Reagent type (species) or resource | Designation | Source or reference | Identifiers | Additional information |
|---|---|---|---|---|
| Gene (RNAseq data of hagfishes *Eptatretus goslinei*, *Eptatretus stoutii*) | Under BioProject PRJNA896978 at https://www.ncbi.nlm.nih.gov/sra | | | |
| Biological sample (hagfishes: *Eptatretus goslinei*, *Eptatretus stoutii*) | Wild-captured | | | |
| Software, algorithm | R (https://www.r-project.org/) | | | |
| Software, algorithm | Code (https://github.com/plachetzki/ETC_GTC) | | | |

## Animal care and euthanasia

Wild-captured Pacific hagfishes (*E. stoutii*) were housed in a 1000 l tank of chilled artificial seawater (34%, 8°C) at Chapman University, CA, USA. Hagfish were anesthetized using clove oil (200 mg/l) (*McCord et al., 2020*). For euthanasia, hagfish were first anesthetized in 200 mg/l of clove oil and then transferred to a lethal dose of MS-222 (250 mg/l).

## Abundance of ETCs

To quantify the abundance of ETCs and the other two epidermal cells, we sampled cell densities using fixed and stained samples of hagfish skin. First, with a series of transverse cross-sections, we sampled cell abundance along the skin circumference. One Pacific hagfish (body length ~45 cm) was fixed with 3% PBS-buffered paraformaldehyde, and then divided into 10 sections of equal length, exposing 9 transverse cross-sections. Of each cross-section, the anterior portion (~1 cm thickness) was embedded in paraffin wax, sectioned (20 µm thick) and transferred to slides (*Figure 2—figure*

*supplement 1*). The tissues were then stained with hematoxylin and eosin (H&E) following standard procedures (*Bancroft and Gamble, 2008*) and mounted with Permount Mounting Medium (Fisher SP15-100). Digital images were taken for the entire skin section using transmitted light microscopy (40× objective, Zeiss Axio Imager 2).

For each cross-section, the anteroposterior position ($P_{AP}$) was defined as the relative distance from the snout (*Figure 2—figure supplement 1*). Next, we traced the profile of epidermis for one arbitrary side using ImageJ (*Rueden et al., 2017*). The dorsoventral position ($P_{DV}$) was defined as the relative distance from the dorsalmost point ($P_{DV} = 0$; at the dorsal ridge). We then sampled sections of ~1 mm long at each of dorsalmost, ventralmost, and lateral positions. The dorsoventral position of each section was calculated as $P_{DV}=(P_b − P_a)/2$, where $P_a$ and $P_b$ are dorsoventral positions of the two ends. Within each section, we manually recorded the number of cells ($N_{cell}$) and calculated the linear density as $\lambda = N_{cell}/L_{section}$ , where $L_{section} = P_b − P_a$ is the section length. Analyses were performed using custom-written scripts in R (*R Development Core Team, 2013*).

Second, we sampled the area density ($\sigma$) of cells in two freshly euthanized hagfishes. From each hagfish, we collected skin samples (2×2 mm) from the lateral region at three anteroposterior positions (0.2, 0.5, and 0.8). Each skin sample was immediately fixed with 3% PBS-buffered paraformaldehyde (30 min), stained with eosin (~2 min), and washed with 75% ethanol. The skin sample was then transferred to a large coverslip (24×50 mm) with the epidermis facing downward and covered by a smaller coverslip (24×40 mm). Images stacks were then taken with an inverted confocal microscope (Zeiss LSM 980).

## Morphometrics of ETCs and contents

We took image stacks for ETCs on H&E-stained slides using laser confocal microscopy (Zeiss LSM 980 with Airyscan). We sampled the size and area density of granules from cross-sectional confocal images of 17 ETCs. With each cross-section, we manually counted the number of granules and digitized the profile of the granule cluster. We then calculated the area density as $\sigma = N/A$ , where $N$ is number of granules and $A$ is cross-sectional area of the cluster.

We further sampled granules from confocal image stacks taken in the axial direction to assess the variation of granule size. On each slice, we approximated each granule as an ellipse by fitting it with the 'oval' tool in ImageJ. We then summarized the size and density of granules with respect to the axial position (as represented by z-direction) using custom-written R scripts.

## Size and shape of epidermal threads

The helical geometry of threads was sampled from confocal image stacks using ImageJ. We chose helix sections that revolved about an approximately straight central axis for at least three consecutive helical loops. We also checked the thread appearance between stacks to make sure it was approximately aligned with the image plane. We placed paired landmarks on the peaks and valleys on each side of the thread section (*Figure 3B*). Later, with custom-written R scripts, we calculated the centerline of each helix as $p_c =< p_i + p_j >$, where $p_i$ and $p_j$ denote points on each bilateral side of the thread and angle brackets denote average. The mean direction of increase was represented by a vector $v_{inc} = \bar{p_c}$ . The thread diameter ($\phi$) was calculated as $\phi = |p_i − p_j|$ for each pair of landmarks and the mean diameter was calculated for each helix. The pitch angle ($\theta$) was calculated for each half loop as the angle between the centerline and a vector normal to the mean direction of increase. Correspondingly, the helical diameter ($D$) was calculated as $D = |p_c|/\tan\theta$, where $|p_c|$ is the length of helical centerline of a given half loop (*Figure 3B*).

Assuming threads are cylindrical and ETCs are ellipsoidal, the volume fraction occupied by threads within ETCs can be approximated as:

$$\frac{V_T}{V_{ETC}} = \frac{\pi r_T^2 L_T}{\frac{4}{3}\pi r_a r_b^2} \tag{3}$$

where $r_T$ is thread diameter, $L_T$ is thread length measured from scraped samples (*Figure 4*), and $r_a$ and $r_b$ are the major and minor axes of the cell, respectively.

## Epidermis wounds

We examined the products of epidermal abrasion caused by frictional contact and laceration caused by sharp surfaces. To simulate the frictional contact with epidermis and collect the products, we

scraped the epidermis of anesthetized hagfishes using a glass coverslip (18×18 mm). In each trial, we oriented the coverslip at an ~45° contact angle to the hagfish skin and scraped along the lateral side for a linear distance of <5 cm. Next, the coverslip was carefully placed onto a glass slide (*Figure 4— figure supplement 1*). The samples were then observed with an upright compound microscope using transmitted light and DIC optics (Zeiss Axio Imager 2) and images were captured with a digital camera (Axiocam 506; 2752×2208 pixels). For free threads, we took individual images with 20× or 40× objective lenses and later stitched them using Adobe Photoshop.

To observe wounded epidermis, we introduced shallow wounds with a scalpel on euthanized hagfishes. We then excised a 2×2 mm skin sample and placed each on a large coverslip (24×50 mm) with the epidermis facing down. The samples were fixed with 4% PBS-buffered paraformaldehyde (~20 min), stained with eosin (~5 min), and washed with 75% ethanol. To minimize disruption, the samples were maintained on the coverslip throughout the staining process. We washed the samples by slightly tilting the coverslip and dropping 75% ethanol from the higher end, with paper towel collecting the liquid at the bottom. We then took images of the samples using confocal microscopy (Zeiss LSM 980 with Airyscan).

## Phylogenetic and comparative transcriptome analyses

Transcriptome assemblies were constructed using Trinity (*Grabherr et al., 2011*) using RNAseq datasets from three replicates of skin and slime gland tissues for *E. goslinei* and *E. stoutii*. Resulting assemblies were filtered using cd-hit and a -c 0.98 parameter setting. Reduced transcriptome assemblies were then translated to protein sequences using Transdecoder (*Grabherr et al., 2011*). Concurrently, reads from the replicate RNAseq datasets were mapped onto the assemblies using Salmon (*Patro et al., 2017*) and differential gene expression analyses were conducted using the Fisher's exact test implemented in EdgeR (*Robinson et al., 2010*) with p-value cutoff of 0.05.

To reduce heterozygosity, comparative transcriptome analyses were conducted using data from a single individuals of *E. stoutii* and *E. goslinei* (*Mincarone et al., 2021*). Because of this, the diversity of thread transcripts identified from a single individual could correspond to prominently expressed loci, alleles, splice-products, and combinations therein. We screened publicly available (Ensembl v. 69; *Cunningham et al., 2022*) coding sequence data from *E. burgeri* but did not detect sequences with homology to either α or γ. The lack of α and γ sequences in the *E. burgeri* genome may be a consequence of chromosome elimination, which has been shown to be prevalent in hagfishes (*Nakai et al., 1995*). While questions on the genetics of the α and γ thread diversity will become clear once more complete genomic resources for hagfish become available, the starkness of the expression differences between α and γ transcripts is notable. While we had no criteria that transcripts be differentially expressed between skin and slime gland for inclusion in our analysis, all α and γ transcripts with expression greater than 50 transcripts per million (TPM) for each replicate were significantly differentially expressed (*Figure 8*).

Database searching and phylogenetic analyses were conducted using the following approach. First, a BLAST (*Altschul et al., 1990*) database was prepared that included protein models from the genomes of *Petromyzon marinus*, *Callorhinchus milli*, and *Danio rerio*, and the translated protein models derived from the transcriptome assemblies of *E. stoutii* and *E. goslinei*. BLAST (*Altschul et al., 1990*) was conducted using α and γ thread sequences (*Koch et al., 1995*) as queries in separate analyses using a low stringency e value of 0.0001 while retaining up to 30 sequences per species. Fewer than 30 sequences were recovered for each species indicating that our searches were exhaustive at this stringency. The resulting sequences were aligned using the LINSI setting in MAFTT (*Katoh and Standley, 2013*), phylogenetic analyses were conducted under the best fit model in IQ-TREE (*Nguyen et al., 2015*), which in both cases was LG+I+G+F. *Danio* sequences in the resulting trees were annotated, including information on expression domains, using UNIPROT (*Wang et al., 2021*) and rooted with distant intermediate filament outgroups. For the α phylogeny outgroups included glial acidic fibrillary protein and other neuronally expressed intermediate filament loci. For the γ phylogeny outgroups included desmins and other distantly related skin expressed keratins. Trees shown in *Figure 8* were pruned from these larger analyses and consisted of approximately full-length sequences that were greater than 300 amino acids in length and that had an expression of TPM >50. See *Figure 8—figure supplements 1–2*. Bioinformatic and statistical code, quantification data, raw trees, and multiple sequence alignments are available at https://github.com/plachetzki/

ETC_GTC (*Zeng, 2023* copy archived at swh:1:rev:beb18f387410b3e823ff9600b687b6518cff31b9).
Raw RNAseq data are available under BioProject PRJNA896978 at https://www.ncbi.nlm.nih.gov/sra.

## Fibrosity of defensive slime

We developed an empirical model to assess the abundance of GTCs in slime glands and exudate. We found there was no significant difference in the aspect ratio (AR; i.e., the ratio of major axis to minor axis) between full glands and newly emptied glands which simplified our calculations of ejected exudate volume. Using image data from a previous study (*Schorno et al., 2018*), we found that newly emptied glands are ~30% smaller in both major and minor axes than full glands (full glands: major axis $\Phi_a$ = 3.51±0.25 mm, minor axis $\Phi_b$ = 2.45±0.15 mm, $N$=11; newly emptied glands: $\Phi_a$ = 2.47±0.25 mm; $\Phi_b$ = 1.71±0.17 mm, $N$=10; means ± SD; *Figure 7—figure supplement 1*). The gland AR was 1.43±0.09 for full glands and 1.45±0.13 for newly emptied glands. The slime glands were then modeled as ellipsoids with a mean AR = 1.44 (see below).

For the total number of slime glands, we dissected one Pacific hagfish (*E. stoutii*; body length, 48 cm), and counted the glands from under the skin. We found a total of 81 glands on the left side and 82 glands on the right side. A total of 163 glands was then used in assessing GTC abundance and productivity.

For the number of GTCs in full glands, we assumed that GTCs are distributed evenly within the glands, which allowed us to estimate the total based on cross-sectional images. For the full glands, we approximated the total number of GTCs ($N_{GTC}$) based on the area density ($\sigma_{GTC}$) and gland volume ($V_G$) as $N_{GTC} = \sigma_{GTC}^{1.5} V_G$ , where $V_G = \frac{4}{3} \pi r_a r_b^2$ and $r_a = 0.5\Phi_a$ and $r_b = 0.5\Phi_b$ are major and minor radius, respectively. Applying mean values of $\sigma_{GTC}$ and gland dimensions from literature (*Supplementary file 1A*), we found there are ~19,300 GTCs in each full gland. With 163 full glands, the total number of GTCs is ~3.15 × 10⁶, which is ~26% of the total number of ETCs in the epidermis (~1.23 × 10⁷, with the hagfish simplified as a cylinder of 45 cm in length and 20 cm in diameter).

For the total number of GTCs ejected to seawater, with mean values of $\sigma_{GTC}$ and gland dimensions from newly emptied glands, we calculate that ~4100 GTCs remain in an emptied gland after ejection. Subtracting the number of remaining GTCs from the total in full glands, ~15,200 GTCs are ejected per gland (*Supplementary file 1A*). The volume of ejected exudate can be calculated as the volumetric difference between full gland and newly emptied gland as: $V_{[ejected]} = V_{G[full]} - V_{G[emptied]}$ . We found 9.4 mm³ exudate was ejected by each gland.

Similarly, the volume of ejected mucous vesicles can be approximated. Using the area proportions ($\delta$) of gland mucous cells in full and newly emptied glands (*Schorno et al., 2018*) and by assuming even distribution of gland mucous cells within slime glands, we calculated the volume of vesicles as: $V_{GMC} = \delta_{GMC} V_G$ for both full and newly emptied glands. The ejected volume of gland mucous cells was then calculated as $V_{GMC[ejected]} = V_{GMC[Full]} - V_{GMC[Emptied]}$ .

In this study, the 'fibrosity index' represents the ratio of total thread length in a material to the total volume (i.e., the combined volume of mucus and seawater in fully deployed defensive slime). To approximate the fibrosity index of defensive slime, we used (1) the total length of GTC threads ejected by one slime gland and (2) an approximation of the total volume of mucus and seawater mixed with these threads, as derived below.

With ejected exudate volume $V_{[ejected]}$ = 9.37 × 10⁻³ ml per gland and exudate density $\rho$ ~ 1 g/ml (a conservative estimate based on *Fudge et al., 2005*), we estimated the weight of ejected exudate to be $W = \rho V_{[ejected]}$ ~9.37 × 10⁻³ g. Also, the combined w/v concentration of thread and mucus is 0.004% in fully-deployed slime (*Fudge et al., 2005*), the volume of seawater mixed with the exudate from a single gland can then be estimated as: $V_{SW} = V_{[ejected]}$/0.004% = 9.37 × 10⁻³ / 0.004% = 234.25 ml. Next, the total volume of liquid in fully-deployed slime is then $V_S = V_{SW} + V_{[ejected]} = \approx$ 2.34×10⁵ mm³. Lastly, the fibrosity of defensive slime was then calculated as:

$$r_F = \frac{L_T N_{GTC}}{V_S} \tag{4}$$

We found the fibrosity is ~6.5 × 10⁵ mm/mm³ for unmixed exudate and ~12 mm/mm³ for fully deployed defensive slime (*Supplementary file 1B*; *Figure 7—figure supplement 1*), which shows that the exudate is diluted 5.5×10⁴ times and that the fully deployed defensive slime is ~800 times less fibrous than epidermal slime.

## Acknowledgements

We thank Andrew Lowe for logistical help and thank Richard Wassersug for comments. This study was supported by NSF grants IOS-1755397 to DF and IOS-1755337 to DP.

## Additional information

### Funding

| Funder | Grant reference number | Author |
| --- | --- | --- |
| National Science Foundation | IOS-1755397 | Douglas Fudge |
| National Science Foundation | IOS-1755337 | David C Plachetzki |

The funders had no role in study design, data collection and interpretation, or the decision to submit the work for publication.

### Author contributions

Yu Zeng, Conceptualization, Data curation, Software, Formal analysis, Supervision, Validation, Investigation, Visualization, Methodology, Writing – original draft, Project administration, Writing – review and editing; David C Plachetzki, Conceptualization, Resources, Data curation, Software, Formal analysis, Supervision, Funding acquisition, Validation, Investigation, Visualization, Methodology, Writing – original draft, Project administration, Writing – review and editing, Conducted the molecular and phylogenetic analyses; Kristen Nieders, Formal analysis, Investigation, Methodology, Collected morphological data of hagfish epidermal threads; Hannah Campbell, Formal analysis, Investigation, Methodology, Collected morphological data of hagfish epidermal threads; Marissa Cartee, Data curation, Software, Formal analysis, Conducted the molecular and phylogenetic analyses; M Sabrina Pankey, Data curation, Software, Formal analysis, Investigation, Visualization, Conducted the molecular and phylogenetic analyses; Kennedy Guillen, Formal analysis, Collected and analyzed morphological data of hagfish epidermal thread cells; Douglas Fudge, Conceptualization, Resources, Data curation, Formal analysis, Supervision, Funding acquisition, Validation, Investigation, Visualization, Methodology, Writing – original draft, Project administration, Writing – review and editing

### Author ORCIDs

Yu Zeng http://orcid.org/0000-0002-2651-227X
Marissa Cartee http://orcid.org/0000-0001-6434-1750
M Sabrina Pankey http://orcid.org/0000-0002-7061-9613

### Decision letter and Author response

Decision letter https://doi.org/10.7554/eLife.81405.sa1
Author response https://doi.org/10.7554/eLife.81405.sa2

## Additional files

### Supplementary files

• MDAR checklist

• Supplementary file 1. Supplementary table.
 (A) Morphological parameters of slime glands of Pacific Hagfish. (B) Comparison of fibrosity between defensive slime and epidermal slime.

### Data availability

All data generated or analysed during this study are included in the manuscript and supporting file. Source Data files have been provided for Figures 2, 3, 4 and 7. Raw RNAseq data are available at https://www.ncbi.nlm.nih.gov/bioproject/?term=PRJNA497829. Bioinformatic and

statistical code is available at https://github.com/plachetzki/ETC_GTC, (copy archived at swh:1:rev:beb18f387410b3e823ff9600b687b6518cff31b9).

The following dataset was generated:

| Author(s) | Year | Dataset title | Dataset URL | Database and Identifier |
|---|---|---|---|---|
| Plachetzki D | 2022 | Raw RNAseq data | https://www.ncbi.nlm.nih.gov/bioproject/PRJNA497829 | NCBI BioProject, PRJNA497829 |

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
