## [Editor Report]

The study is a careful investigation of the physical properties of hagfish slime and the underlying cellular framework that enables this extraordinary evolutionary innovation. It is a careful and detailed measurement with clear images. The revised manuscript provides a better contextualizing of the findings as a broader biological question, including the evolution of functional novelty, the adaptive processes, and the links between genetic and phenotypic evolution. The transcriptome analysis of several species further supports the evolutionary model. Therefore, this paper provides solid evidence for a unique and important view of the slime and should be of interest to those working on hagfish and on these secretions.

---

## [Decision Letter]

**Decision letter after peer review:**

Thank you for submitting your article "Epidermal threads reveal the origin of hagfish slime" for consideration by *eLife*. Your article has been reviewed by these reviewers, and the evaluation has been overseen by myself, Claude Desplan, as the reviewing editor and as the Senior Editor. The following individual involved in the review of your submission has agreed to reveal their identity: Omer Gokcumen (Reviewer #2).

Essential revisions:

The reviewers found the work to be of potentially great interest and were enthusiastic about the entire topic. However, they have some significant comments that must be addressed before the paper can be accepted.

1) As per reviewer #1, you need to contextualize better the issue and the evolution of the slime.

2) You must present your paper with the various possible mechanisms that led to the evolution of the slime: As it stands, the paper aims to test only one hypothesis without addressing any alternative. Being consistent with your model is not a sufficient demonstration of its validity.

3) More specifically, your genomic analysis must be better developed, taking into consideration the state of the genome of the hagfish that might be missing numerous genes.

*Reviewer #1 (Recommendations for the authors):*

1) Hypothesis testing

– Expand and reorganize Discussion 3.5 "Implications for the origin of hagfish slime" with clarity in mind. It would be a good idea to summarize the origin scenario in discrete steps and then explain for each step what selection mechanism may have acted on it.

– Any detectable/documented variation in slime composition and release mechanism/dynamics among LIVING HAGFISHES that could fit and strengthen one or more of the steps?

– Along with the selective regime, I think it's important to consider and spell out the trade-off. For example, rupture-based release posited for an early evolutionary stage has an obvious trade-off (skin has to be damaged to release threads), which would then favor more spontaneous release.

– It will enhance the hypothesis testing if there is any alternate hypothesis (cellular origins and/or selection mechanisms). For example, Glover et al. (2012) Proc. B. reported amino acid transport across hagfish skin and described some similarities between hagfish epidermis and intestinal epithelium. They have since framed fish skin in the evolution of transport epithelia. This might provide a viable alternative hypothesis to test against (=Could any of the steps for the origin scenario be reframed with the skin as transport epithelia?). I don't expect the thread cells could evolve in the intestinal epithelium, but the mucous cells? Are any data (histological or transcriptomic) available for the hagfish intestine?

– Or else, it would be a good idea to consider making predictions about what might falsify the origin scenario.

2) Transcriptomic and phylogenetic analyses

As I outlined in my public comment, I am not convinced that these analyses "demonstrate" the authors' scenario. Simplified trees in Figure 5 do not accurately reflect the topologies of the data supplements. They do suggest gene duplications, but splice variants and ontogenetic genome reorganization are real issues with cyclostomes. The current genome assembly of Petromyzon is considered incomplete. Are these hagfish transcripts vetted against the draft genome (unpublished but worked on by multiple groups)? Did the authors check the sequence disparity?

My recommendations here are:

– Expand Results 2.7 "Epidermal threads are ancestral to slime threads" and explain Figure 5 in detail. Particularly, richly describe rationales for the proposed polarity.

– It is not clear to me whether the authors argue that polymer gene duplications were a prerequisite to the origin of slime or that the duplication events followed and elaborated it. I would like to know what they predict here, and how those predictions are correlated with the steps of their origin scenario in the Discussion.

– This is completely my bias, but, intuitively, I'm more used to seeing vertical volcano plots (counts per million on y-axis, fold changes on x).

All in all, the authors try to solve an enduring, unanswered question in biology and present a strong narrative and high-quality data that support it. These critical comments do not diminish any part of their achievement. They deserve congratulations.

*Reviewer #2 (Recommendations for the authors):*

Overall, I like the detailed dissection of the slime properties and the cellular origins of how eels produce this amazing substance. I have a couple of major concerns, however:

1. The contextualization: I have had a hard time figuring out the main question answered here. As it stands, it reads as a very descriptive paper. More specifically, as a non-expert on this subject, I could not really orient myself on what is known and what is not known about hagfish slime. More generally, I could not find the links to broader biological questions. Is the evolution of hagfish slime unique? How is this compared to other related substances (as they started doing in Figure 3B), and is there a phylogenetic/evolutionary trend associated with these relationships (convergence, similar ecological threads)? I am an evolutionary biologist, but I can think about other paths in bioengineering, cytology, biochemistry, etc. where broader questions can be posed with this amazing model system that the authors so diligently measured.

2. The evolutionary/transcriptomic analysis: I could not follow how the authors determined the specific genes in the transcriptomic analysis. I understand that keratins are interesting, but there may many other genes and unexpected ways in which this phenotype may be regulated. This section needs a lot more details to be convincing with regards to the identified genes and their relevance to the slime phenotype. I am also not convinced with the conclusions, based on a single gene, about the evolutionary origins of this phenotype. In this regard, I do not think that the results necessarily support the conclusions.

3. The writing suffers from excessive use of acronyms. Some parts, such as the measurement methodologies, should go to the Methods section. Overall the writing can be shortened, tightened, and simplified, especially considering the general audience of *eLife*.

---

## [Author Response]

Reviewer #1 (Recommendations for the authors):1) Hypothesis testing– Expand and reorganize Discussion 3.5 "Implications for the origin of hagfish slime" with clarity in mind. It would be a good idea to summarize the origin scenario in discrete steps and then explain for each step what selection mechanism may have acted on it.

Thank you for the suggestion. We re-wrote Discussion section 3.5 and described discrete steps as a possible evolutionary trajectory to accompany the schematic in Figure 9B.

– Any detectable/documented variation in slime composition and release mechanism/dynamics among LIVING HAGFISHES that could fit and strengthen one or more of the steps?

All extant species of hagfishes have slime glands, and while there is certainly variability in some subtle aspects such as skein and thread dimensions (see Zeng et al. 2021; https://doi.org/10.1016/j.cub.2021.08.066), anecdotally it appears that all species possess the same fundamental components such as mucus, threads, and active ejection via striated muscle contraction. In other words, there does not appear to be variability among living hagfishes in traits that reflects the major transitions we propose here. This suggests that all living hagfishes are descended from a common ancestor that possessed slime glands with all the essential features of modern hagfishes. Very little is known about variability in hagfishes regarding the thread and mucous cells in their epidermis.

– Along with the selective regime, I think it's important to consider and spell out the trade-off. For example, rupture-based release posited for an early evolutionary stage has an obvious trade-off (skin has to be damaged to release threads), which would then favor more spontaneous release.

We agree. Trade-offs are certainly relevant, but we also think that timing and opportunity are also important considerations. Even if active release would have been more effective than bite-induced damage and release, it is difficult to imagine how this could have evolved de novo without some intermediate stage. We believe that defensive secretions triggered by bites and skin damage was the first step on the evolutionary path that led to hagfish slime, and that this trait is retained in a modified form in modern hagfish skin. Furthermore, the holocrine mode of exudate release from slime glands, in which cells rupture on their way out of the gland, may be the result of the cell trauma origins of defensive secretions from the skin.

– It will enhance the hypothesis testing if there is any alternate hypothesis (cellular origins and/or selection mechanisms). For example, Glover et al. (2012) Proc. B. reported amino acid transport across hagfish skin and described some similarities between hagfish epidermis and intestinal epithelium. They have since framed fish skin in the evolution of transport epithelia. This might provide a viable alternative hypothesis to test against (=Could any of the steps for the origin scenario be reframed with the skin as transport epithelia?). I don't expect the thread cells could evolve in the intestinal epithelium, but the mucous cells? Are any data (histological or transcriptomic) available for the hagfish intestine?

Thank you for this thoughtful comment. There is one alternative that has been discussed in the literature, and that is a cloacal origin of the slime glands (Fudge et al. 2015. 10.1146/annurev-biochem-060614-034048). We have added a discussion of this possibility in the Discussion (Line 369).

Regarding the transport function of hagfish skin, we did not mention this because we can’t think of a mechanism where this function might predispose the skin to secreting defensive cells and cell products.

– Or else, it would be a good idea to consider making predictions about what might falsify the origin scenario.

Thank you for this suggestion. The epidermal origins hypothesis would be endangered by evidence that GTCs arose before ETCs. Our transcriptomic and gene phylogeny data suggest the opposite, that the thread proteins in ETCs evolved first and then diversified within GTCs in the slime glands. If the pattern had been the opposite, with ancestral thread genes expressed in slime glands and more derived genes in the epidermis, this would cast serious doubt on the epidermal origin hypothesis. The revised text in the Discussion now makes this point clear.

2) Transcriptomic and phylogenetic analysesAs I outlined in my public comment, I am not convinced that these analyses "demonstrate" the authors' scenario. Simplified trees in Figure 5 do not accurately reflect the topologies of the data supplements. They do suggest gene duplications, but splice variants and ontogenetic genome reorganization are real issues with cyclostomes. The current genome assembly of Petromyzon is considered incomplete. Are these hagfish transcripts vetted against the draft genome (unpublished but worked on by multiple groups)? Did the authors check the sequence disparity?

The analyses presented in the revision now include transcriptome data from an additional hagfish species, *Eptatretus stouti*. In addition, the phylogenetic analyses of α and γ thread biopolymers are better annotated in the revision and, we feel, clearer. The reviewer is correct to point out that in the original submission, our presentation of the argument that slime gland expressed loci are most likely descended by duplication of skin expressed loci was insufficient. We have clarified our presentation and in the main text elaborated on an additional line of support that the sister gene families for both α and γ thread biopolymers are also keratins that are predominately expressed in skin, further bolstering the argument that skin expression is the likely ancestral state for α and γ thread biopolymers. We have also revised most of the text dealing with these results and now include information on expression domains in the gene trees, clarifying this point specifically. The reviewer is also correct to point out that in the original submission, the gene trees reported in Figure 5 (now Fig8) did not accurately reflect the gene trees from which they were drawn. This was due to a rooting error that occurred in the production of the original figure. Our revised figure includes a better depiction of results from phylogenetic analyses of α and γ thread biopolymers. Additionally, the analyses of the additional hagfish species, *Eptatretus stoutii,* in the revised manuscript enhances the strength of our previous arguments and we observe a similar phylogenetic/expression pattern in both species, where a low diversity of skin expressed α and γ thread biopolymer transcripts are sister to a larger diversity of slime gland expressed transcripts. With the addition of this species, we can also observe patterns in the gene trees suggestive of hagfish-specific orthologs among the transcripts analyzed. As described in the original and revised manuscripts, and is now clearer in the supplements, all the α and γ thread biopolymer transcripts we report in Figure 8 (previously Figure 5) are highly expressed. We do recover a few transcripts with low expression (less than 50 TPM), which we interpret as splice products, however these do not change our overall interpretation of results in any way. Splice products are reported in the gene tree supplements but not in the main figure. While we did not have access to unpublished, alternative assemblies for *Petromyzon*, we have no reason to suspect any problem with using these data. Our analyses do not assume that any of the genomic resources utilized in the study are complete. In addition, the gene sequences from *Petromyzon* that were captured by our analytical approach are consistent with our conclusions from the phylogenetic analyses of α and γ thread biopolymers. Moreover, the presence of skin-expressed α and γ thread biopolymers from the well-annotated, polished, genome of the zebrafish, *Danio rerio*, is sufficient to draw our conclusion, irrespective of the *Petromyzon* data. In the future, better quality genome resources for cyclostomes will help distinguish genetic loci from splice products, but none of our conclusions presented are dependent on such knowledge.

My recommendations here are:– Expand Results 2.7 "Epidermal threads are ancestral to slime threads" and explain Figure 5 in detail. Particularly, richly describe rationales for the proposed polarity.

We expanded Results section 2.7 and added more detailed textual and graphical explanation for Figure 5 (now Figure 8).

– It is not clear to me whether the authors argue that polymer gene duplications were a prerequisite to the origin of slime or that the duplication events followed and elaborated it. I would like to know what they predict here, and how those predictions are correlated with the steps of their origin scenario in the Discussion.

We expanded our discussion of how gene duplication may have been involved in the origin of slime and slime threads in Results section 2.7 and Discussion section 3.4.

– This is completely my bias, but, intuitively, I'm more used to seeing vertical volcano plots (counts per million on y-axis, fold changes on x).

Volcano plots would be appropriate here, however, given the graphical design of the new Figure 8, MA plots, as presented, fit better. The information included in both styles of plots are the same.

All in all, the authors try to solve an enduring, unanswered question in biology and present a strong narrative and high-quality data that support it. These critical comments do not diminish any part of their achievement. They deserve congratulations.

Thank you.

Reviewer #2 (Recommendations for the authors):Overall, I like the detailed dissection of the slime properties and the cellular origins of how eels produce this amazing substance. I have a couple of major concerns, however:1. The contextualization: I have had a hard time figuring out the main question answered here. As it stands, it reads as a very descriptive paper. More specifically, as a non-expert on this subject, I could not really orient myself on what is known and what is not known about hagfish slime. More generally, I could not find the links to broader biological questions. Is the evolution of hagfish slime unique? How is this compared to other related substances (as they started doing in Figure 3B), and is there a phylogenetic/evolutionary trend associated with these relationships (convergence, similar ecological threads)? I am an evolutionary biologist, but I can think about other paths in bioengineering, cytology, biochemistry, etc. where broader questions can be posed with this amazing model system that the authors so diligently measured.

Thank you for these insightful comments. In the revised manuscript, we focused on addressing the question “where did hagfish slime and slime glands come from”? To answer this question, we tested the hypothesis “hagfish slime and slime glands are originated from epidermis” using morphological, functional and molecular data. We have also rewritten the Introduction to frame our study in a broader ecological and evolutionary context.

2. The evolutionary/transcriptomic analysis: I could not follow how the authors determined the specific genes in the transcriptomic analysis. I understand that keratins are interesting, but there may many other genes and unexpected ways in which this phenotype may be regulated. This section needs a lot more details to be convincing with regards to the identified genes and their relevance to the slime phenotype. I am also not convinced with the conclusions, based on a single gene, about the evolutionary origins of this phenotype. In this regard, I do not think that the results necessarily support the conclusions.

We added some text explaining our focus on thread genes α and γ, which are responsible for the major structural proteins within slime threads and were identified as intermediate filament genes and then sequenced in the mid 1990s. Because the slime threads are what makes hagfish slime differ from other mucus-based secretions, we focused on α and γ in our analysis. We are currently conducting a more thorough analysis of genes expressed in hagfish skin and slime glands, but that is beyond the scope of this manuscript. We also have rewritten parts of the Discussion where we more clearly explain how the transcriptomic data support an epidermal origin of the slime glands.

3. The writing suffers from excessive use of acronyms. Some parts, such as the measurement methodologies, should go to the Methods section. Overall the writing can be shortened, tightened, and simplified, especially considering the general audience of eLife.

To improve the readability, we restricted the use of acronyms to the two main cell types of interests (GTCs and ETCs). We also moved Equation 1 (volume of epidermal threads) to Methods and only kept the essential information on how our data were collected. We have also done a thorough revision for clarity and concision.